# A somatic genetic clock for clonal species

Lei Yu ®[1,10], Jessie Renton[2,10], Agata Burian ®[3], Marina Khachaturyan ®[1,4], Till Bayer ®[1], Jonne Kotta ®[5], John J. Stachowicz ®[6], Katherine DuBois[6], Iliana B. Baums ®[7,8,9], Benjamin Werner ®[2] ✉ & Thorsten B. H. Reusch ®[1] ✉

Age and longevity are key parameters for demography and life-history evolution of organisms. In clonal species, a widespread life history among animals, plants, macroalgae and fungi, the sexually produced offspring (genet) grows indeterminately by producing iterative modules, or ramets, and so obscure their age. Here we present a novel molecular clock based on the accumulation of fixed somatic genetic variation that segregates among ramets. Using a stochastic model, we demonstrate that the accumulation of fixed somatic genetic variation will approach linearity after a lag phase, and is determined by the mitotic mutation rate, without direct dependence on asexual generation time. The lag phase decreased with lower stem cell population size, number of founder cells for the formation of new modules, and the ratio of symmetric versus asymmetric cell divisions. We calibrated the somatic genetic clock on cultivated eelgrass *Zostera marina* genets (4 and 17 years respectively). In a global data set of 20 eelgrass populations, genet ages were up to 1,403 years. The somatic genetic clock is applicable to any multicellular clonal species where the number of founder cells is small, opening novel research avenues to study longevity and, hence, demography and population dynamics of clonal species.

Clonal reproduction is the process of generating (potentially) physically independent multicellular organisms (that is, ramets *sensu*[1]) via mitosis, a widespread life history among animals, plants, macroalgae and fungi[2]. Starting from a single zygote, multipotent somatic cells proliferate to form new ramets via branching or budding, often becoming physiologically independent after a few years when severing from the parental tissue. All modules or ramets stemming from that single zygote represent a genet (or clone). Often, the contribution of sexual and clonal reproduction to local population structure varies among species and localities[3–5], resulting in asexual populations of ramets that are nested within the 'classical' population of genets[2,6]. Coral, algae, seagrass or poplar genets, for example, can reach considerable size and, therefore, age with linear extents of >1 km (refs. 7–11). The apparent persistence and resilience of asexual ramet populations is astonishing in light of the considerable temporal and spatial variation they may experience over their lifetimes despite little genetic variation (but see refs. 10,12) and raises questions about these species' adaptability in a rapidly changing climate[13].

As a key parameter to evaluate this persistence, genet age/longevity has been inherently difficult to estimate, in particular, when biomass tracing back to an individual's origin is not preserved, as is the case in non-woody plants[14]. For example, a small genet is not necessarily young if episodes of ramet mortality reduced its size in the past. To estimate genet age via molecular genetic methods, somatic genetic variation (SoGV) segregating among ramets has previously been used. However, those attempts lacked resolution, as the SoGV could be estimated at only a few marker loci[9,15].

[1]GEOMAR Helmholtz-Center for Ocean Research Kiel, Marine Evolutionary Ecology, Kiel, Germany. [2]Evolutionary Dynamics Group, Centre for Cancer Genomics and Computational Biology, Barts Cancer Institute, Queen Mary University of London, London, UK. [3]Institute of Biology, Biotechnology and Environmental Protection, University of Silesia in Katowice, Katowice, Poland. [4]Institute of General Microbiology, Kiel University, Kiel, Germany. [5]Estonian Marine Institute, University of Tartu, Tallinn, Estonia. [6]Department of Evolution and Ecology, University of California, Davis, CA, USA. [7]Helmholtz Institute for Functional Marine Biodiversity, University of Oldenburg, Oldenburg, Germany. [8]Alfred Wegener Institute, Helmholtz-Centre for Polar and Marine Research (AWI), Bremerhaven, Germany. [9]Institute for Chemistry and Biology of the Marine Environment (ICBM), School of Mathematics and Science, Carl von Ossietzky Universität Oldenburg, Oldenburg, Germany. [10]These authors contributed equally: Lei Yu, Jessie Renton. ✉e-mail: b.werner@qmul.ac.uk; treusch@geomar.de

In this Article, we present a novel approach to estimate genet age on the basis of a somatic genetic clock that uses complete genome information of the focal species. Molecular clocks were initially developed for species-level phylogenies and rely on the neutral theory of molecular evolution[16]. Fixed neutral mutations within species accumulate at a constant rate equal to the rate of spontaneous mutations[17], and thus, genetic differences between species increase with absolute time[18,19]. If the mutation rate can be derived on the basis of calibration points such as fossil evidence, clock estimates can be extended to phylogenetically related clades[20]. Recently, fixation of SoGV was demonstrated in clonal species through a process of somatic genetic drift[12]. During genet growth via new ramet formation, somatic mutations become fixed in the descendant ramets, essentially because only a few pluripotent cells of the proliferating tissue are recruited to form the new module or ramet[12,21,22] (Fig. 1 and Supplementary Fig. 1). Here, we built upon these findings and introduce the somatic genetic clock that uses the rate of genome-wide, asexual fixation of alleles to estimate the extent of differentiation between the founder and descendant ramets of a genet. In doing so, we can infer the time to the least common ancestor of multiple or pairs of ramets, here the zygote, and derive a 'somatic genetic clock' that permits the precise ageing of large plant clones (genets) and, possibly, other clonal animal, macroalgal or fungal species.

## Results

### A generic somatic genetic clock in clonal species revealed by modelling and simulations

To estimate the time over which fixed SoGV accumulates and segregates under clonal growth, we developed a stochastic, agent-based model of a generic clonal organism that comprises a collection of modules, adapted from population genetics models of cancer evolution[23] (Methods). Within this model, a module is simplified to the stem cell population of a single ramet (all somatic cells are derived from stem cells and, thus, can be ignored). Cells and modules are subject to stochastic update events including cell division, death and the formation of new modules, with new Poisson-distributed mutations occurring at each cell division. We considered a range of scenarios with different types of stem cell division (symmetric versus asymmetric) that characterize stem cell dynamics in clonal species. Specifically, we compared different (founder) stem cell pool sizes, and varying rates and mechanisms for forming new modules (branching versus splitting), attempting to capture possible life history variation in clonal species across the tree of life (Fig. 2). We found that, given sufficient time, any scenario would converge to a constant accumulation rate of fixed SoGV, and thus, the number of fixed SoGV would increase linearly with clonal age (Fig. 3 and Supplementary Figs. 2 and 3) as required for a useful molecular clock.

The accumulation rate of fixed SoGV was determined solely by the mutation rate per cell per site per year. While the module formation rate ($r$) does not directly impact the accumulation rate of fixed SoGV (Fig. 1), it can have a small indirect effect by altering the mutation rate, either as a result of stochasticity or because of different effective mutation rates during homeostasis and growth. This effect is small (Supplementary Fig. 4), and we consider it negligible for biologically relevant parameter ranges. The relative constancy despite different module formation rates, that is, asexual generation times, is equivalent to the classical molecular clock being dependent only on mutation rate and not sexual generation time[17,19,24].

We next explored the duration of the lag phase before linearity is reached and found that it depended upon the size of the stem cell pool per module ($N$), the number of founder stem cells that are recruited to form new modules ($N_0$), the ratio of symmetric versus asymmetric cell division, the rate of stem cell division ($b$), the rate at which new modules are formed ($r$) and whether they are formed by branching or splitting (Fig. 3a and Supplementary Figs. 5 and 6). Module formation via a small number of founder stem cells (small $N_0$) reached a linear equilibrium

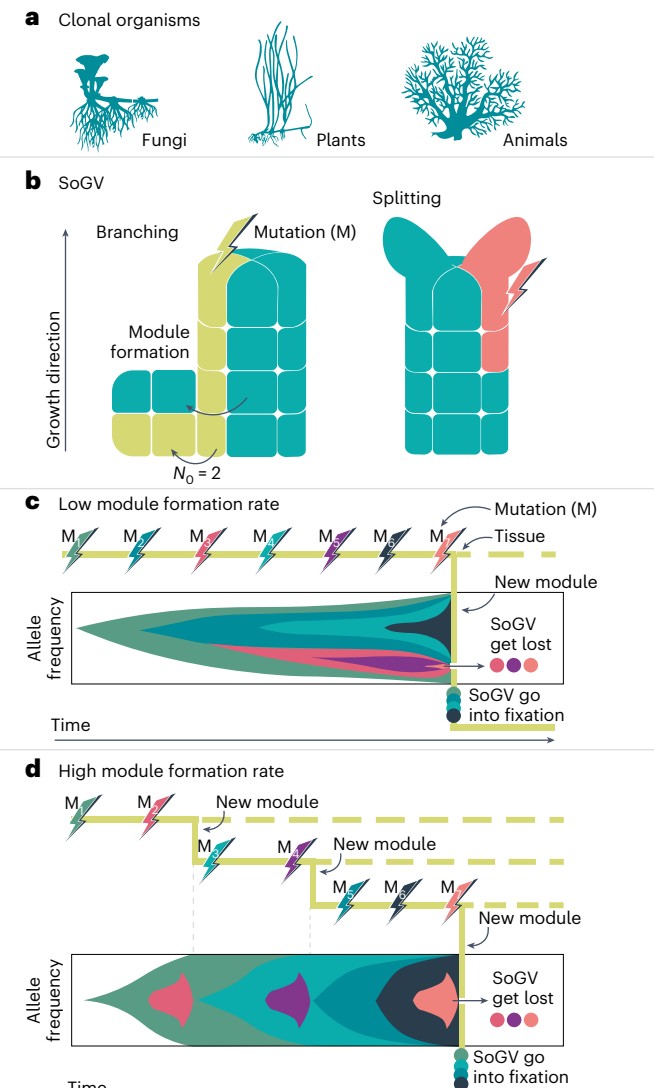

**Fig. 1 | Dynamics of SoGV in generic clonal organisms. a**, Multicellular clonal species exist across the tree of life. **b**, Allele frequency change of SoGV due to the formation of new modules by branching or splitting. A new module is initiated either directly by the stem cells (that is, splitting) or by the daughter cells of the stem cells (that is, branching). Splitting reduces the size of the original stem cell population, while branching leaves the original cell population untouched. During the formation of new modules, the cell population undergoes a genetic bottleneck. **c,d**, The accumulation rate of fixed SoGV is independent of module formation rate. The tree topology depicts a module undergoing (multiple) module formation events, where the dashed line and the solid line represent the original module and the new module respectively. New mutations (M) occur at a constant rate, and only mutations in the new modules are depicted (with a different colour). For each timepoint, the vertical length of the colours represents the frequency of the SoGV within the module. Clonal dynamics in a single module (solid line in tree structure) are depicted as a Muller plot that shows the nested allele frequency of SoGV over time. The frequency of SoGV changes during module formation events, due to the bottleneck. Eventually, SoGVs are either fixed or lost. Under low module formation rate (**c**), fixation events are rare. Thus, many SoGVs have accumulated in the intervening time and are fixed simultaneously. Under high module formation rate (**d**), fixation events occur more frequently, but with fewer SoGVs fixed at each branching event.

fast for both branching and splitting (Fig. 3a and Supplementary Fig. 6). The duration of the lag phase increased substantially for a large number of founder cells and/or solely asymmetric stem cell divisions. Fixation of SoGV occurs due to the repeated formation of new modules, during

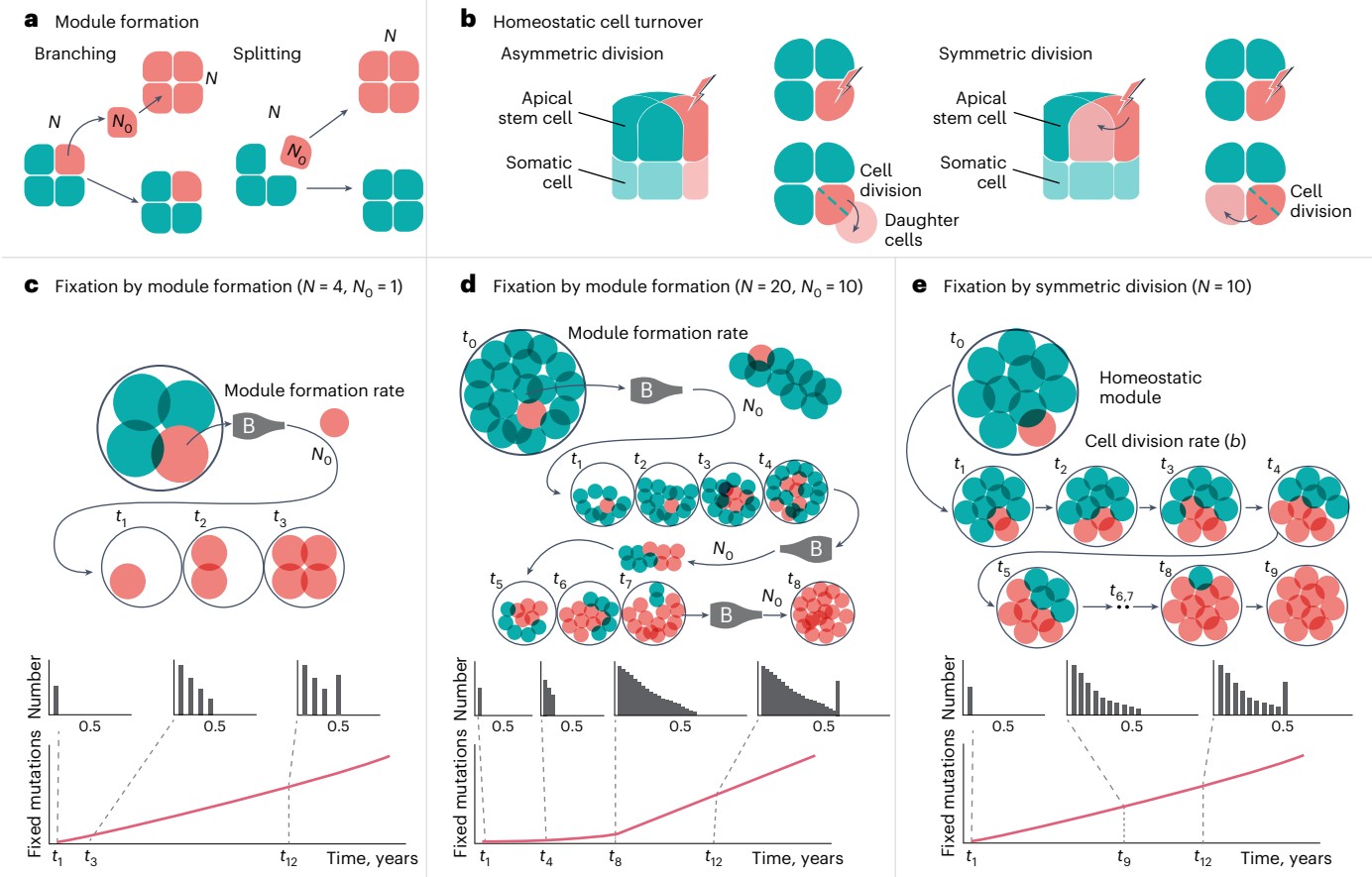

**Fig. 2 | Processes determining fixation rates of SoGV in a generic clonal organism.** Module representation was reduced to the stem cell compartment of each ramet within the clonal organism. **a**, New module formation occurs by branching or splitting of homeostatic modules, during which $N_0$ cells are selected from a parent module to form the founding population of a new module. For module branching, these are copied (leaving the parent unchanged), while for module splitting, they are removed from the parent. Growth by cell division is then implemented so that all modules return to homeostatic size $N$. **b**, Cell turnover in homeostatic modules occurs by asymmetric or symmetric division of stem cells. The dashed lines depict dividing stem cells that produce two daughter cells. After asymmetric division, one daughter cell remains in the module and the other differentiates (leaving the stem cell compartment). Under symmetric division, one daughter cell replaces one of the other stem cells (which is assumed to differentiate), and thus, both daughter cells remain in the module. **c–e**, The

frequency of a new mutation within the stem cell population ($N$) is $1/(2 \times N)$ for diploid species. This frequency will change during clonal proliferation. If not lost by drift, persistent mutations can be visualized on the basis of their frequencies relative to the total number of chromosomes in the stem cell population, that is, $1/(2 \times N), 2/(2 \times N), …, N/(2 \times N)$. A frequency of $N/(2 \times N) = 0.5$ means the mutation is shared by all the stem cells, reaching the fixation state (that is, fixed SoGV). The number of fixed SoGV accumulates linearly in modules once an equilibrium is reached. SoGVs become fixed in modules by repeated bottlenecks (depicted as a bottle labelled 'B') induced by module formation (that is, module size reduces from $N$ to $N_0$, then regrows). The time period of the nonlinear phase is shorter for smaller $N$ and $N_0$ (**c**) compared with larger $N$ and $N_0$ (**d**). SoGVs become fixed in modules during homeostasis in the case of symmetric division (**e**). This is similar to classic population genetic models (that is, a Moran process), and the time period of the nonlinear phase increases with $N$.

which the population of cells that form the module undergoes a bottleneck (Fig. 2). Additionally, fixation can occur due to homeostatic cell turnover within the module if, and only if, there is symmetric cell division, while this cannot occur for purely asymmetric divisions.

Next, we estimated the conditional fixation times for different clonal species' life histories. Assuming asymmetric cell division, fixation occurs only due to repeated module formation, which can be represented as a modified Wright–Fisher process. We derive the conditional fixation times, which are approximately $4N_0 (1 − N_0/N)/r$ (equation (1)) for module splitting and $4N_0/\left(1 − N_0^2/N^2\right) r$ (equation (2)) for module branching (see Supplementary Note 1.3 for the derivation using a diffusion approximation). Thus, fixation times may be decreased by reducing $N_0$, even when $N$ is large (Supplementary Fig. 5). For symmetric cell division, fixation due to homeostatic cell turnover usually dominates, because the cell division rate $b$ is greater than the module formation rate $r$. The conditional fixation time is therefore better represented by a Moran process, and is approximately $N/b$

(equation (3), ref. 25). The conditional fixation time can be considered as a lower bound on the lag phase to reach the equilibrium accumulation rate of fixed SoGV. Thus, these equations indicate the absolute timescale over which the somatic genetic clock is applicable for different species life histories.

Finally, we also considered additional complications with respect to the developmental mode of the clonal organisms. Under (1) stochastic quiescence, homeostatic modules move in (and out) of a quiescent state with a fixed rate; while under (2) seasonal quiescence, all modules become quiescent during a winter period (Supplementary Note 1). We also consider the possibility that mutations may occur during the cell lifetime, as well as at cell division. To this end, we introduce a time-dependent mutation rate $\xi$ in addition to the per-cell mutation rate $\mu$ (Supplementary Note 1 and Supplementary Fig. 7). The lag phase before linearity is increased for both quiescence regimes (Supplementary Fig. 7a–c), indicating that the average rate of module formation across the population is lowered. However, linearity is still reached in all cases.

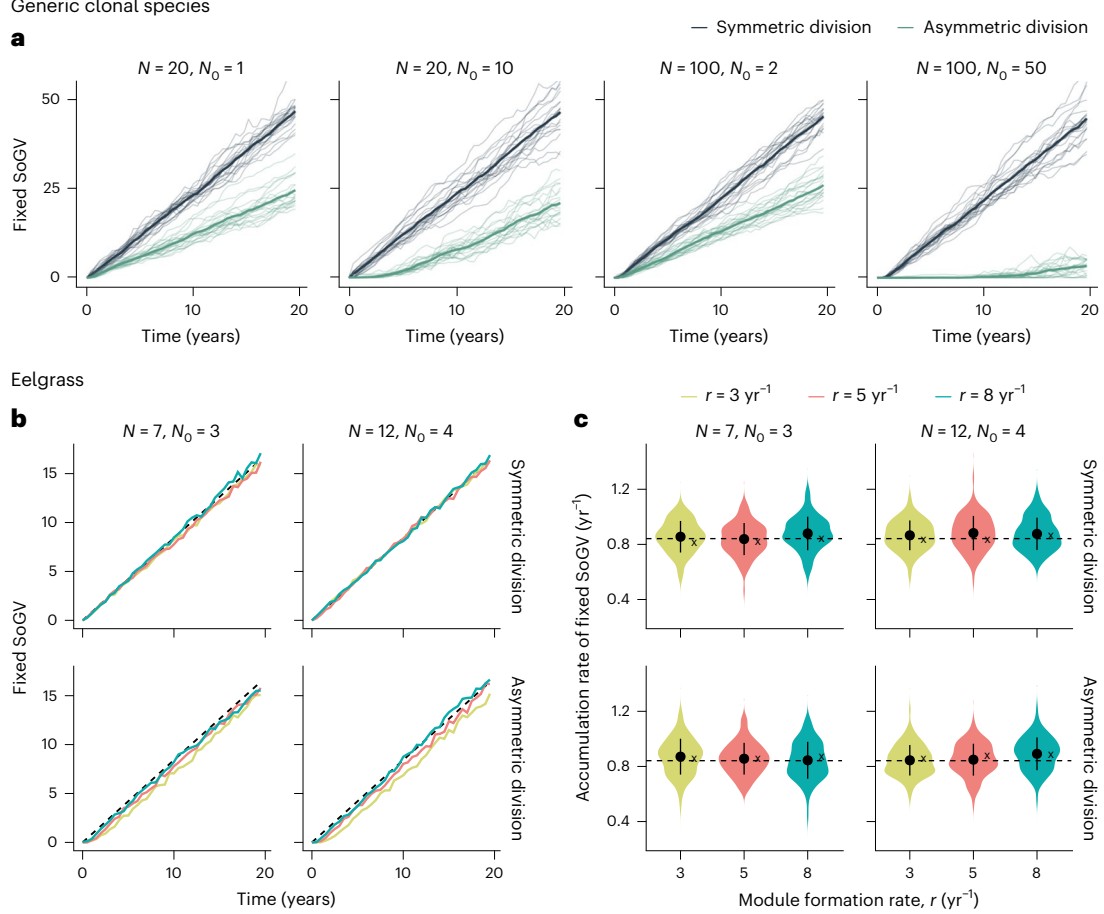

**Fig. 3 | Agent-based model predictions for the accumulation of fixed somatic mutations via somatic genetic drift. a**, Model for a generic clonal species. Simulations are shown for a range of model regimes, with new modules formed by module branching. After a lag phase, the rate of accumulation of fixed SoGV reaches an equilibrium and becomes linear. However, in some cases ($N$ = 100; $N_0$ = 50, asymmetric division), the lag phase is long. Thin lines, over all modules of a single genet; thick line, over all 20 genets. Chosen parameters: $\mu$ = 0.01, $b$ = 122 yr$^{-1}$, $r$ = 5 yr$^{-1}$, $Z$ = 100 (symmetric and asymmetric update events occur at rate $b$). **b**, Model parametrization for eelgrass (Supplementary Figs. 7–9), demonstrating that the equilibrium rate of accumulation of fixed SoGV is reached quickly. In eelgrass, new modules are formed by branching. In each simulation, the mean fixed mutations are calculated at each timepoint from a random sample of ten modules, and data are means over ten simulations (dashed line: $\mu b$ (approximation of the mutation rate per cell per year)) with the accumulation rate of fixed SoGV estimated from simulated modules at two timepoints mimicking the experimental methodology (4 years: three genets with two sampled modules per clone, and 17 years: two clones with five and six sampled modules, respectively; Methods) **c**, Mean fixed SoGV is calculated for each eelgrass genet, and the accumulation rate is then estimated by linear regression. Violin plots depict mean (±1 s.d.) and density kernel distribution of 100 replicate simulations (dashed line: $\mu b$; '$x$' gives estimated mean accumulation rate of fixed SoGV by linear regression on 100 simulated 200-year-old clones). Parameters: $\mu$ = 0.0069, $b$ = 122 yr$^{-1}$, $Z$ = 1,000 (symmetric update events occur at a rate $b/2$, asymmetric update events at rate $b$, Table 1).

## Application of the somatic genetic clock in a seagrass

We then applied the somatic genetic clock to the seagrass *Zostera marina* (eelgrass), an emerging model for evolution in clonal plants. We first examined the structure of the shoot apical meristem (SAM) containing a population of stem cells in higher plants[26] via laser confocal microscopy. We were interested in evidence for SAM stratification that determines the spread of SoGV across tissues[27], along with the likely number of stem cells ($N$) and module founder cells ($N_0$), as well as the stem cell division mode (symmetric or asymmetric) (Supplementary Note 2). We found that the SAM was organized into one-layered L1 (tunica) and underlying L2 (corpus) as in many other monocotyledonous plant species (Supplementary Fig. 8a). No periclinal cell division in L1 was observed during the formation of axillary meristems, indicating a stable boundary between L1 and L2 (Supplementary Fig. 8b–d). In contrast, frequent periclinal cell divisions in L1 were observed during the formation of leaves, which suggested that L1 mostly or exclusively contributed to leaves (Supplementary Fig. 9). A likely number of L1 stem cells is between 7 and 12 with possible both asymmetric and symmetric cell division modes (Supplementary Fig. 10). From this population, about three or four stem cells give rise to cells that form a new module.

Next, we addressed how a SoGV can become fixed throughout the entire tissue of a new module despite meristem stratification. Indeed, we find clear allele fixation at $f$ = 0.5 in variant frequency diagrams (for example, >7,000 with $f$ = 0.5; ref. 12 and Supplementary Fig. 11). Although shoot meristems are generally stratified in *Z. marina* as in other angiosperms[28] (Supplementary Fig. 8), it cannot be excluded that infrequent periclinal cell divisions occur in the L1 (ref. 29) leading to SoGV fixation in all tissues. Note that leaf tissues that are derived exclusively from L1 (Supplementary Fig. 9) were predominating in the sample used for bulk sequencing. We thus continued by simplifying the fixation dynamics by assuming a one-layer case, enabling the application of our model of a generic clonal organism to eelgrass. However, assuming that cell growth and division frequency is similar across layers[30], the model can be applied to any cell layer and derived organs in stratified meristems[31].

We parametrized the model for eelgrass and focused on the most likely range with $N = 7–12$ and $N_0 = 3–4$ (Fig. 3) but also considered more extreme scenarios ranging from the strongest ($N = 7$, $N_0 = 1$) to the weakest ($N = 12$, $N_0 = 6$) intensity of somatic genetic drift, in combination with branching rates 3–8 yr$^{-1}$ (refs. 32,33). The accumulation rate of fixed SoGV remained similar (Fig. 3b and Supplementary Figs. 12a and 13), indicating that mutation accumulation on the size of the SAM and rate of asexual reproduction was negligible.

Using equations (2) and (3), we estimated the conditional fixation times for novel mutations under asymmetric and symmetric cell division, respectively, within an eelgrass clone. For the most likely parameter range, these gave reasonable lower and upper bounds of 2 years ($N = 7$, $N_0 = 3$, $r = 8$ yr$^{-1}$) and 6 years ($N = 12$, $N_0 = 4$, $r = 3$ yr$^{-1}$) for asymmetric cell division, and 0.05 years ($N = 7$, $b = 122$ yr$^{-1}$) and 0.1 years ($N = 12$, $b = 122$ yr$^{-1}$) for symmetric division. This suggests that a constant accumulation rate required for the somatic genetic clock will be reached relatively fast in eelgrass, in the order of years or even months. This is verified by our simulations (Fig. 3b) in which we observe very small lag times ($\lesssim 1$ year) for symmetric cell division. For asymmetric cell division, it took longer to reach an equilibrium, with the time increasing for smaller module formation rate ($r$) and larger (founder) module size. However, the lag times still appeared in the order of years, rather than decades.

## Calibration of the somatic genetic clock

Next, two long-term cultivation experiments with *Z. marina* genets of known age (4 and 17 years, respectively) allowed for a calibration of the somatic genetic clock. Owing to statistical noise in estimating the true allele frequency via mapped reads at a given locus, differentiating between mosaic and fixed SoGV is inherently difficult. Hence, we developed the variable 'Variant Read Frequency 50 (R$_x$)' (hereafter VRF50(R$_x$)) as a proxy for the number of fixed SoGV in ramet 'R$_x$' relative to the founder of the genet (Methods and Supplementary Fig. 13). The mean VRF50(R$_x$) of a ramet population can be used as estimator for its age, that is, the time since founding by a parent genet or a zygote. In order to calibrate the somatic genetic clock for *Z. marina*, genets of known ages (4 and 17 years) were deep-sequenced (~900× and ~80× for the genets aged 4 and 17 years, respectively) to calculate the accumulation rate of VRF50(R$_x$). The role of sequencing depth and type of mutation caller were also examined (see Methods for details; Supplementary Tables 1–3). The mean VRF50(R$_x$) and the age of a genet were used to fit a linear model (Fig. 4a; $y = 0.5044x – 1.4641$, adjusted $R^2$: 0.9483, $P < 0.001$). Accordingly, we find a rate of fixed mutation accumulation of $4.6 \times 10^{-9}$ per year per site, similar to estimates in *Arabidopsis*[34].

To verify that our data could be used to accurately calibrate the clock, we re-created the sampling strategy for both timepoints, that is, 4 and 17 years, by simulation and estimated the accumulation rate of fixed SoGV (Fig. 3c and Supplementary Fig. 12b). Considering data from 100 simulations for each parameter setting, we observe similar estimated rates in all cases. The difference between the mean estimated rate and 'true' rate was between 0.1% and 8%, where the maximum difference is for the most extreme case ($N = 12$, $N_0 = 6$, $r = 3$ yr$^{-1}$). The standard deviation (s.d.) for each parameter setting ranged between 0.10 and 0.15 (mutations per year). As this was similar in magnitude to other sources of error, we consider that our calibration genets with known ages of 4 years and 17 years can be safely used for calibration. However, increasing the number of samples (more genets at given age/a higher range of genet ages) would probably reduce the error resulting from sampling.

## Age estimation of 15 globally distributed *Z. marina* genets

We then used the calibrated somatic genetic clock to estimate the age of eelgrass genets in a worldwide data set[35] (Fig. 4b and Supplementary Data 1). Among the 15 genets with two or more ramets sampled, most were <40 years old (Fig. 4c), while 4 attained >270 years (Fig. 4d), 1 in

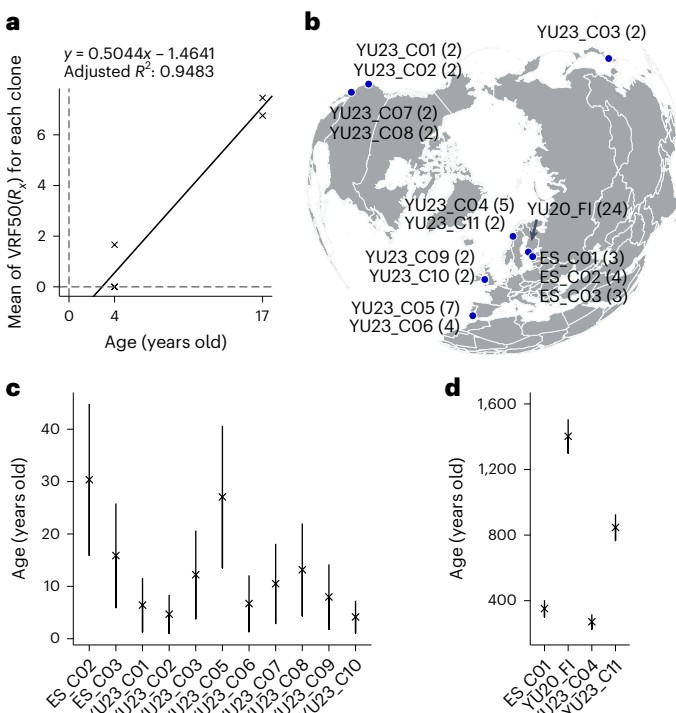

**Fig. 4 | Estimating the age of globally distributed eelgrass (*Z. marina*) clonal lineages based on the somatic genetic clock. a**, The accumulation rate of the VRF50(R$_x$). Each data point represents one clonal lineage. Two of the three 4-year-old clonal lineages show mean VRF50(R$_x$) of 0, and the data points overlap with each other. **b**, The location of the 15 globally distributed eelgrass genets. The number in parentheses indicates the number of ramets for each corresponding genet. **c**, Age estimates for genets <100 years of age. The data are presented as mean values ± 95% confidence interval. The sample size (that is, the number of independent ramets) for each clone can be found in Supplementary Data 1. **d**, Age estimates for the four oldest genets. The detailed information for age estimates is available in Supplementary Data 1.

Estonia (352 years), 2 in Norway (271 and 847 years) and 1 in Finland (1,403 years). All genets >270 years of age were located in higher latitudes (>50° N) in the North Atlantic, indicating that marginal populations were more likely to maintain old genets[4,11,36] and supporting the established geographic parthenogenesis pattern[37]. Although the evolutionary history in the Pacific is much longer than that in the Atlantic[35], Pacific eelgrass genets were young (<40 years). In addition, the old clonal lineages were distributed in the locations that were recolonized by glacial refugia after the Last Glacial Maximum, indicating that clonal reproduction is a particularly successful reproductive mode to rapidly colonize newly opened areas[4]. Note that age estimates based on spatial extent would have been misleading, as genets with small spatial extent were found to be >300 years old. For example, while genet ES_C01 in Estonia contained only three ramets spread over ~20 m (Supplementary Fig. 15), it was estimated to be 352 years old based on the somatic genetic clock.

We also examined the mutational spectra of the four oldest genets detected here to six sexually segregating North Atlantic populations to identify possible differences between germline and asexually generated, mitotic mutations (Supplementary Fig. 16). Both mutational patterns show the well-known predominance of transitions over transversions in plants[34,38]. In particular, G:C → A:T transitions contributed 61 ± 4% and 66 ± 3% (mean ± 1 s.d.; n = 6 and 4, respectively) of all single-nucleotide polymorphism (SNPs) in asexual versus sexual populations, respectively, regardless of their trinucleotide context (Supplementary Fig. 16a,b).

## Discussion

We present a somatic genetic clock that permits precise age estimates of genets in clonally growing plants, and possibly, many clonal animal, fungal and macroalgal species. The duration of the lag time before the DNA-sequence-based somatic genetic clock approaches linearity decreases for fewer stem cells and founder stem cells; for symmetric rather than asymmetric cell divisions; and for increased rates of new module formation. Hence, an application of the somatic genetic clock is most accurate for estimating clonal age if the stem cell population size $N$ is small and new module formation happens through a small founder cell population $N_0$ as realized in plant SAMs. In organisms that asexually reproduce through budding, time to linearity will depend on the number of cells contributing to the new bud. Conversely, marine invertebrates or algae that propagate asexually through fission will have an exceedingly long lag time, as essentially half of all body cells comprise the founder cell population $N_0$. By applying our analytical results (equations (1)–(3)), we are able to estimate the timescale over which the somatic genetic clock is applicable for any given organism.

Once linearity is reached, the rate of the somatic genetic clock is constant across module formation rates and, thus, asexual generation times, which is the hallmark of a valid molecular clock. Similar to the rate constancy despite different generation times in species-level phylogenies[17,24], under a higher module formation rate, fewer mutations are fixed by any single module formation event, but the total number of module formation events is higher (Fig. 1), and vice versa. Our proposed clock is analogous to mitotic evolution in non-modular species, such as humans, specifically the emergence of genetic heterogeneity among healthy and cancerous human somatic tissues within an individual[39,40]. Somatic mutations accumulate linearly with age in human stem cells and fixate at a constant rate locally in spatially constrained stem cell populations, for example, in colon crypts or skin[41,42]. Similarly, we find that the number of fixed SoGV between founder and descendant ramets also accumulates at a constant rate.

Currently, we cannot distinguish mutations resulting from DNA replication errors during mitotic divisions from those occurring outside cell division. Indeed, recent studies suggest that somatic mutations can also accumulate with age in both plants and animals[41,43]. This indicates that, independently of cell division dynamics, other factors such as ultraviolet radiation, transposons or insufficient DNA repair systems could also increase the accumulation of mutations over time. A comparison of mutational spectra (Supplementary Fig. 16a,b) does not suggest that the frequency of a type of transitions commonly associated with environmental stress in plants is increased under long-term clonal growth. Even if this was the case in other species, it would rather enhance the validity of our somatic genetic clock, as it decouples somatic mutation accumulation from developmental processes.

The stem cell population dynamics during module formation are currently unknown for most clonal species other than angiosperms. The latter are complicated due to stem cell stratification into layers[28]. However, even under these circumstances, the somatic genetic clock can be applied when either the sampled tissue is dominated by one meristematic layer (as is the case in eelgrass *Z. marina*) or when descendant tissues of a certain stem cell population can be clearly distinguished among the adult plant organs[31].

Our findings on fixation processes will also apply to an evolutionary epigenetic clock that was recently described for self-fertilizing and clonally reproducing plants[44]. This clock uses the much faster accumulation of neutral gene body (de)methylations of cytosine nucleotides. As an additional step, the identification of genomic regions with clock-like behaviour of (de)methylation is required[44]. The somatic genetic clock proposed here is complementary and will be best suited for slightly longer time intervals of >10 years to potentially tens of thousands of years, and where methylation data are unavailable. Here, we provide the theoretical foundation why both, the somatic genetic clock

### Table 1 | Model parameters

| Symbol | Description |
|--------|-------------|
| $b$ | Cell division rate during growth |
| $\lambda$ | Symmetric division (Moran) rate during homeostasis |
| $\gamma$ | Asymmetric division rate during homeostasis |
| $r$ | Module formation rate |
| $N$ | Homeostatic module size (number of stem cells) |
| $N_0$ | Initial module size (number of founder stem cells) |
| $Z$ | Population size (maximum number of modules) |
| $\mu$ | Mutation rate per cell per division |

and the evolutionary epigenetic clock[44], are ultimately determined by mutation rate, as is the case for general molecular clocks[24].

Some of the analogies of our modelled and observed temporal dynamics with classic population genetics are instructive. In our study, the stem cell population size, and the time period between two adjacent branching events, correspond to the population size $N_e$, and generation time in classic population genetics, respectively. Due to the usually large $N_e$ (>100) in combination with genetic exchange among lineages, classic molecular clocks are limited to macro-evolutionary timescales ($\sim 10^5$–$10^8$ years). However, the stem cell population size in plants is extremely small (for example, 7–12 for eelgrass, but for other angiosperms often only 3–4 (ref. 26) or even only 2 in some species of ferns[45]), and module formation events often occur multiple times per year, which makes somatic genetic clock solid for recent timescales. Note that the time until stem cell populations are 'saturated' with standing genetic variation, resulting from novel mutations, increases with population size $N_e$, similar to time lags required for a population to reach mutation–drift equilibrium in population genetics[46].

With increasing availability of full genome data at the population level, our study provides an achievable and accurate method for estimating the age of clonal plants and, possibly, other clonal species in the animal, macroalgal and fungal kingdom[2]. It opens multiple new research avenues to model the demography, resilience and evolution of the many species that are facultatively clonal, and where direct and precise ageing information was previously unavailable.

## Methods

### Simulating fixed mutation accumulation in a clonal organism

We implemented a stochastic, agent-based model of a clonal organism, adapted from population genetics models of cancer evolution[23]. The organism is represented as a population of modules that grows to a fixed size $Z$ by producing new modules via module splitting or branching (Table 1). Modules consist of stem cells and have different dynamics depending on whether they are in growth or homeostasis. During the growth phase, the module grows by cell division, which is implemented by a stochastic pure-birth process with rate $b$. Once the module reaches size $N$, it enters homeostasis. Cell divisions are coupled with cell deaths, so that the population size remains constant. This is done either by implementing an asymmetric update (a cell divides producing only one progeny) or a symmetric update (a cell divides producing two progeny and another cell is removed from the module). This symmetric update corresponds to a Moran process. Dividing cells acquire novel, Poisson-distributed mutations with mean $\mu$.

Homeostatic modules produce new modules at rate $r$. This is done by module splitting or module branching. For module splitting, the parent module donates $N_0$ cells to the new child module. Both parent and child modules then re-enter the growth phase. For module branching, $N_0$ cells are sampled without replacement from the parent module and then copied to form the child module that enters a growth phase. The parent module is unchanged. If the population of modules has reached maximum size $Z$, a randomly selected module is killed whenever a new

module is formed to keep the population size constant. The simulation is implemented using a Gillespie algorithm[47]:

1. Initialize the simulation with one module that is formed of a single cell, $t = 0$.
2. Calculate the transition rates for all transitions:
   a. Cell division in a growing module: $r_a = bn_{growth}$
   b. Symmetric division in a homeostatic module: $r_b = \lambda N Z_{homeostatic}$
   c. Asymmetric division in a homeostatic module: $r_c = \gamma N Z_{homeostatic}$
   d. New module formation: $r_d = r Z_{homeostatic}$

Here, $n_{growth}$ is the total number of cells in growing modules and $Z_{homeostatic}$ is the number of homeostatic modules. We set $\lambda = b$ (or $b/2$), $\gamma = 0$ for purely symmetric division and $\lambda = 0$, $\gamma = b$ for purely asymmetric division.

3. Transition $i$ is chosen with probability $r_i / (r_a + r_b + r_c + r_d)$. If a cell division occurs during any transition, the newly divided cells acquire $M \sim$ Poisson ($\mu$) novel mutations. Possible transitions are:
   a. Choose a cell to divide uniformly at random from all cells in growing modules.
   b. Choose a homeostatic module, uniformly at random. From that module, choose a cell to divide and a different cell to remove, uniformly at random (Moran update).
   c. Choose a homeostatic module, uniformly at random. From that module, select a cell to divide, also uniformly at random. One progeny cell remains in the module, and the other is removed (asymmetric division).
   d. Choose a homeostatic module uniformly at random to be the parent module, and if $Z = Z_{max}$, choose a second module to die. A new module is formed from the parent module by (i) splitting or (ii) branching. First, select $N_0$ cells from parent module without replacement, then,
   (i) Module branching: copy them to form a new module, leaving the parent module unchanged.
   (ii) Module splitting: remove them from the parent module to form a new module.
4. Update the time $t' = t + \delta t$, where
   $\delta t \sim$ Exponential $(1/(r_a + r_b + r_c + r_d))$.
5. Repeat steps 2–4 until $t = T_{max}$.

Data are generated at discrete time steps for the number of fixed SoGV in each module.

## Shoot apex preparation and imaging in laser confocal microscope

*Z. marina* plants collected in Falckenstein, Kiel Fjord (54.392° N, 10.192° E) were kept at 8–12 °C temperature and 150 μmol quanta s$^{-1}$ m$^{-2}$ light intensity in 800-litre indoor wave tanks, the 'Zosteratron', receiving ambient Baltic seawater while rooted in ambient sediment (12 cm deep), with an intake pipe approximately 10 km distant from the collection site. The plants were then either moved immediately to room temperature for 2–3 days and imaged, or the temperature was slowly raised to 16 °C temperature for 7 days to induce growth before imaging. We used the plants at the vegetative phase of development.

For the imaging in the laser confocal microscope, plants were dissected in filtered seawater using tweezers and fine medical needles under a stereo-microscope (Nikon) so that all leaf primordia covering the SAM were cut off. Isolated shoot apices (SAMs with the youngest leaf primordia) and axillary meristems were fixed and prepared for the imaging according to ClearSee-based clearing method[48]. Isolated apices were fixed with 4% paraformaldehyde dissolved in the phosphate-buffered saline buffer (pH 6.9–7.0 adjusted with HCl) for at least 2 h (at the first hour, under vacuum). Apices were washed twice in the phosphate-buffered saline buffer for at least 2 min, and incubated for 7–18 days in the ClearSee solution (2% urea, 10% xylitol and 15% sodium deoxycholate) at room temperature with gentle stirring. The ClearSee solution was changed every 1–2 days. Cell walls were stained with 0.05% Fluorescent Brightener 28 (FB, Sigma) dissolved in the ClearSee solution for at least 30 min, rinsed in the ClearSee solution and washed in fresh water for 1–2 min.

For the imaging, the apices were mounted in small containers filled with 5% of low-melting-point agarose and kept in fresh water. The imaging was performed using an upright confocal laser-scanning microscope (Leica TCS SP8) with long-working distance water-immersion 40× objective. For the FB, excitation and emission 405 nm and 425-475 nm wavelengths were used, respectively. Images were collected at 12 bits. Scanning speed was set at 400 Hz with 512 × 512 or 1,024 × 1,024-pixel frames, zoom at 0.75–2.0 and z-step at 0.3–0.8 μm. The pinhole was set at 1AU (airy units).

## Image processing and analysis

Original confocal z-stack images (LIF) were converted using Fiji (https://fiji.sc) to TIFF files, which were then processed with the MorphoGraphX (MGX) v.2.0.1 (ref. 49) to obtain top or site views and optical sections. To analyse the structure of apices, a series of optical 2–4-μm-thick sections were performed parallel and perpendicular to the SAM major axis (longitudinal and transverse sections, respectively). Developmental stages of leaf primordia were estimated on the basis of optical transverse sections through the apex. The p1 is the youngest primordium apparent as a bulge at the SAM surface. The successive stages were numbered in ascending order (p2, p3 and so on; Supplementary Figs. 8–10).

To estimate the number of stem cells at the SAM surface, cell clones were analysed (Supplementary Fig. 10). Cell clones (usually containing 4–16 cells) were recognized on the basis of the history of cell divisions at projections of SAM anticlinal cell walls. Specifically, the FB signal was projected in the MGX software from the defined depth (0–3 μm) onto the SAM surface. At these projections, the signal is the most intense in newly formed cell walls corresponding to most recent cell divisions (higher-order divisions). The signal in the oldest cell walls (regarded as clone borders) is the weakest due to a furrow formed over time between descendant cells.

## Parameterizing the model for eelgrass

The modelling for eelgrass was focused on layer L1. New module formation was implemented by module branching, reflecting the fact that in eelgrass the new SAM is not directly derived from the stem cells (Supplementary Note 2). The following parameter range was used: $b = 122$ yr$^{-1}$ (ref. 26); $r = 3$–8 yr$^{-1}$ (refs. 32,33); $N = 7$–12; $N_0 = 1$–7; $Z = 1,000$; $\mu = 0.0069$. Both symmetric and asymmetric cell division were considered by setting $\lambda = b/2$, $\gamma = 0$ or $\lambda = 0$, $\gamma = b$, respectively.

## Eelgrass genets for calibration cultured in the lab

**Four-year-old eelgrass calibration genets.** Three small eelgrass patches, consisting of 17–25 leaf shoots were collected in April 2019 from an eelgrass meadow in Kiel, Germany (Falckenstein, 54.392° N, 10.192° E). To confirm clonal identity, each patch was carefully excavated by divers to examine rhizome connections and additionally genotyped with nine microsatellite loci[50]. In the Baltic Sea, seeds germinate in March or April, while plants become mature at the end of year one. The observed number of shoots can be obtained by branching in the second year. Hence, we infer that the collected eelgrass patches were probably founded by seeds that germinated in 2017 and started branching in 2018. Plants were tagged, planted into 40-litre plastic boxes to a sediment height of 15 cm and placed into 800-litre wave tanks with flow-through ambient Baltic seawater at the GEOMAR Helmholtz Center for Ocean Research Kiel, the 'Zosteratron'. Leaf shoot number was regularly reduced to allow clones to regrow and branch. In 2022, 3 years after start of the cultivation, one leaf shoot from each of boxes

was selected and resequenced to ~900× coverage using a Novaseq 6000 S4 platform (paired end reads of 150 bp). The estimated time between tissue collection and seedling emergence was 4 years (3 years in the lab + 1 year in the field). Sequence data are available at BioProject no. PRJNA1025927, accession no. SRR26321801-804 and SRR26321811-812.

**Seventeen-year-old eelgrass calibration genets.** Data are from a whole-genome resequencing of two eelgrass genets with a known age of 17 years (ref. 51). Each genet was initiated by a single shoot collected from Bodega Harbour, California, in July 2004. Before sample collection plants had been kept for 17 years in large, 300-litre outdoor flow-through mesocosms at Bodega Marine Laboratory under ambient light and temperature conditions[52]. Six and five ramets were collected from each genet for genomic analysis in 2021, respectively. The clone assignment was checked on the basis of shared heterozygosity[51]. Illumina sequencing data are available in the National Center for Biotechnology Information (NCBI) sequence read archive (~80×, BioProject no. PRJNA806459, SRA accession nos. SRR18000159–SRR18000170).

**Sampled eelgrass genets in the field**
**ES_CO1-ES_CO3.** We conducted novel whole-genome resequencing for ten leaf shoots collected from an eelgrass meadow in Estonia (Supplementary Fig. 14). They were chosen from a larger sampling based on microsatellite data that suggested they belong to three genets, containing three, four and three ramets, respectively. This was confirmed by whole-genome SNPs. The clonal lineages were named 'ES_C01' to 'ES_C03' in this study. Data are available in BioProject no. PRJNA1025927, SRA accession nos. SRR26321797–SRR26321810.

**YU2O_FI.** Whole-genome resequencing for 24 ramets of a single large eelgrass genet was conducted in Finland at Ängsö[12]. The next-generation sequencing data are available in the NCBI short read archive (~80×, BioProject no. PRJNA557092, SRA accession nos. SRR9879327–SRR9879353).

**YU23_C01-YU23_C11.** In a large population data set encompassing Pacific and Atlantic sites, 190 ramets from 16 geographic locations were re-sequenced[35], which revealed 11 genets in total that comprised 2–13 ramets. Previously, only one ramet per detected genet was included in subsequent phylogeographic analyses. Here, genets were named 'YU23_C01' to 'YU23_C11', and the respective among-ramet genetic differentiation was used for age determination. Next-generation sequencing data are available in the NCBI short read archive[35].

**Whole-genome resequencing data of new populations**
Bulk DNA of the meristematic region and the basal portions of the leaves was extracted using NucleoSpin Plant II Kit (Macherey-Nagel). DNA concentration was determined using a Qubit Fluorometer (Thermo Fisher Scientific) and Nanodrop Spectrophotometer (Thermo Fisher Scientific), and DNA quality was checked by agarose gel electrophoresis. DNA was sent to Beijing Genomics Institute (Hong Kong) for library construction and sequencing. The libraries were sequenced on either Novaseq 6000 S4 platform (PE150bp) or Hiseq Xten platform (PE150bp).

**Mapping the sequencing data to the reference genome**
We assessed the quality of the raw reads using FastQC v0.11.7 (https://www.bioinformatics.babraham.ac.uk/projects/fastqc/). BBDuk (https://jgi.doe.gov/data-and-tools/software-tools/bbtools/bb-tools-user-guide/bbduk-guide/) was used to remove adapters and for quality filtering according to the following criteria: (1) sequence downstream with quality <20 was trimmed (trimq = 20); (2) reads shorter than 50 bp after trimming were discarded (minlen = 50); (3) reads with average quality below 20 after trimming were discarded (maq = 20). FastQC was used to do a second round of quality check for

the clean reads. Clean reads were then mapped against the *Z. marina* reference genome v2.1 (ref. 53) using BWA-MEM v0.7.17 (ref. 54) with default parameters. The aligned reads were sorted using SAMtools v1.7 (ref. 55), and duplicated reads were marked using MarkDuplicates tool in GATK v4.0.1.2 (ref. 56). Only properly paired reads (0 × 2) with MAPQ of at least 20 (-q 20) were kept using SAMtools.

**Clone assignment check for ramets collected from Estonia**
GATK4 was used to conduct joint SNP calling for the ten eelgrass ramets selected at three sites in Estonia. HaplotypeCaller was used to generate a GVCF format file for each individual, and GenotypeGVCFs was used for SNP calling based on the combined GVCF file from CombineGVCFs. After filtering (GitHub), the shared heterozygosity method[51] was used to verify clonemate pairs that had already been pre-selected by microsatellite genotyping of a larger number of ramets ($n = 10-15$) per site.

**Somatic polymorphism calling and calculation of VRF50($X_1, X_2$)**
Eelgrass (*Z. marina*) is diploid, and ~99.67% of the genome is homozygous. Hence, in most cases, a somatic mutation changes a homozygous to a heterozygous genotype. For SNP detection, the software packages Mutect2 (ref. 57) and Strelka2 (ref. 58) developed originally for cancer mutation calling were used. Both SNP callers compare the 'normal' sample and the 'tumor' sample. Here, SNPs were assumed to represent the ancestral 'normal' case if homozygous for the reference allele, because most novel mutations will turn a homozygous to a heterozygous site. Accordingly, the 'tumor' sample carried the novel alternative allele. For a specific Mutect2/Strelka2 run with $X_1$ as the 'normal' sample and $X_2$ as the 'tumor' sample, we used VRF50($X_1, X_2$) to represent the number of somatic mutations in $X_2$ with a variant read frequency (VRF) ≥0.5. VRF50($X_1, X_2$) was calculated as the number of SNPs meeting the following criteria: (1) the coverage of $X_1 \geq 12$; (2) the coverage of $X_2 \geq 23$; (3) the VRF of $X_1 \leq 0.01$; (4) the VRF of $X_2 \geq 0.50$.

We also examined the role of sequencing depth on the SNP calling results. Two data sets were compared: (1) three ramets of the oldest Finnish clone sequenced to 1,370× depth using a Novaseq platform versus 80× coverage on an Illumina platform (Supplementary Table 1); (2) randomly reducing a 900× data set to 80× coverage for three 4-year-old calibration genets (Supplementary Table 2).

**Analysis of mutational spectra**
Mutational spectra of soma and germline mutations were compared. Mutations were extracted from vcf files after SNP calling and classified according to substitution types and one base up- and downstream context into 96 categories. Graphs were produced with the MutationalPatterns R package. We compared germline mutations within six North Atlantic populations derived from the 11,705 core SNP set from ref. 35 to somatic mutations detected in the four oldest genets identified in this paper (cf. Fig. 4b).

**Calculation of VRF50($R_x$)**
During clonal growth, the fixation of SoGV within all the stem cells leads to substitutions compared with the founder ramet (for the eelgrass case, see Supplementary Fig. 1). We defined $S(R_x)$ to represent the number of the fixed SoGV (that is, substitutions) in the ramet $R_x$ compared with the founder seedling/ramet. By definition, the fixed SoGV have an allele frequency of $f = 0.5$ under diploidy. Based on sequencing data, allele frequency could be estimated by the VRF. In the histogram of VRF, the fixed SoGV form a peak at VRF 0.5 (Supplementary Fig. 1). However, for a normal coverage (<100×), mosaic distribution overlaps with the left-hand part of the fixation distribution. Hence, we focused on only the right-hand part of the fixation distribution, and used VRF50($R_x$) as a proxy for $S(R_x)$, which was the number of the fixed SoGV with a VRF ≥0.5.

After a specific time period from the initiation of the clonal lineage, the number of fixed SoGV in a ramet/module $R_x$, $S(R_x)$, is expected to follow a Poisson distribution, $S(R_x) \sim \text{Poisson}(\lambda)$. For a given $S(R_x)$, the

VRF has equal probability to be >0.5 or <0.5 and, thus, VRF50($R_x$) is assumed to follow a binomial distribution, VRF($R_x$) ~ B(S($R_x$), 0.5). The expectation of VRF50($R_x$) is $0.5\lambda$.

We used VRF50($R_x$)$_{obs}$ to represent the value of VRF50($R_x$) detected from the sequencing data that sufficiently cover a subset of the reference genome. To obtain VRF50($R_x$)$_{obs}$, the most straightforward logic would be to compare the founder ramet/seedling and the target ramet $R_x$. However, the founder did not exist anymore after it had divided into two daughter ramets. Thus, we did an indirect calculation of the VRF50($R_x$)$_{obs}$ (Supplementary Fig. 13). For example, to obtain VRF50(R01)$_{obs}$, each of the other collected ramets of the same clonal lineage was used as the 'normal' sample in SNP calling (Mutect2 or Strelka2), and the maximum value for VRF50(clonemate of R01, R01) was assigned to VRF50(R01)$_{obs}$. Both SNP caller packages Mutect2 and Strelka2 were used to calculate VRF50($R_x$)$_{obs}$ for the clonal lineages with known age, and the results were similar (Supplementary Table 3). For the remainder, we used Mutect2 for comparability with older results[12] and as it seems more conservative.

Note that the sequencing data sufficiently cover only a subset of the genome. To estimate the genome coverage, HaplotypeCaller (GATK4) was run for each ramet using BP_RESOLUTION mode (-ERC BP_RESOLUTION). We then counted the number of the nucleotide sites with coverage ≥23 (that is, Size_e). The average VRF50($R_x$) for a clonal lineage was calculated as (average VRF50($R_x$)$_{obs}$)/(average Size_e) × total genome size. The 95% confidence interval of the average VRF50($R_x$) was estimated on the basis of the Poisson distribution, that is, average VRF50($R_x$) ± 1.96 × sqrt(average VRF50($R_x$)) (Supplementary Data Table 1).

### Estimating the number of mutations and genet age

The average VRF50($R_x$) and the age for the clonal lineages with known age were used to fit a linear model (Fig. 4, $y = 0.5044x - 1.4641$, adjusted $R^2 = 0.9483$, $P < 0.001$), based on which the age of other clones was estimated (Fig. 4 and Supplementary Data Table 1). The number of fixed mutations that had accumulated in a ramet population of a given genet was calculated as 2 × VRF50($R_x$) assuming a symmetric distribution of VRF of fixed somatic SNPs at $f = 0.5$ (Supplementary Data 1).

### Reporting summary

Further information on research design is available in the Nature Portfolio Reporting Summary linked to this article.

## Data availability

All DNA sequence data have been deposited in Genbank (Sequence Read Archive, detailed metadata in Supplementary Data 1). Source data are provided with this paper.

## Code availability

Custom-made scripts and intermediate data steps were deposited on GitHub (https://github.com/leiyu37/SomaticGeneticClock (bioinformatics) and https://github.com/jessierenton/somatic-genetic-clock (modeling)).

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

## Acknowledgements

This work has been funded by the Human Frontiers in Science (HFSP), grant number RGP_0042_2020 to I.B.B., B.W. and T.B.H.R. B.W. is also supported by a Barts Charity Lectureship (grant no. MGU045) and a UKRI Future Leaders Fellowship (grant no. MR/V02342X/1). J.K is supported by the Horizon Europe Programme MARBEFES project (grant no. 101060937). M.K. was supported by a fellowship by the Helmholtz School for Marine Data Science (MarDATA, grant no HIDSS-0005). We thank F. Wendt for maintaining seagrass cultures and M. Timmermans for providing access to the confocal microscopy. We are grateful to S. Landis for assistance with preparing the figures.

## Author contributions

T.B.H.R., I.B.B. and B.W. designed the project and obtained funding. L.Y. and M.K. prepared the DNA samples for sequencing and performed the bioinformatic analysis under supervision of T.B.H.R. J.R. contributed the modelling and simulations along with B.W. A.B. provided histological analyses and microscopic images, T.B. performed the mutational spectra analyses. J.K., J.J.S. and K.D. provided access to field sites and contributed biological material. J.R., L.Y. and T.B.H.R. wrote an initial draft of the manuscript. All authors interpreted the results, and edited and approved the final manuscript.

## Funding

## Competing interests

The authors declare no competing interests.

## Additional information

**Correspondence and requests for materials** should be addressed to Benjamin Werner or Thorsten B. H. Reusch.

# Reporting Summary

## Statistics

For all statistical analyses, confirm that the following items are present in the figure legend, table legend, main text, or Methods section.

| n/a | Confirmed | |
|---|---|---|
| ☐ | ☒ | The exact sample size (*n*) for each experimental group/condition, given as a discrete number and unit of measurement |
| ☒ | ☐ | A statement on whether measurements were taken from distinct samples or whether the same sample was measured repeatedly |
| ☒ | ☐ | The statistical test(s) used AND whether they are one- or two-sided<br>*Only common tests should be described solely by name; describe more complex techniques in the Methods section.* |
| ☒ | ☐ | A description of all covariates tested |
| ☒ | ☐ | A description of any assumptions or corrections, such as tests of normality and adjustment for multiple comparisons |
| ☐ | ☒ | A full description of the statistical parameters including central tendency (e.g. means) or other basic estimates (e.g. regression coefficient) AND variation (e.g. standard deviation) or associated estimates of uncertainty (e.g. confidence intervals) |
| ☐ | ☒ | For null hypothesis testing, the test statistic (e.g. *F*, *t*, *r*) with confidence intervals, effect sizes, degrees of freedom and *P* value noted<br>*Give P values as exact values whenever suitable.* |
| ☒ | ☐ | For Bayesian analysis, information on the choice of priors and Markov chain Monte Carlo settings |
| ☒ | ☐ | For hierarchical and complex designs, identification of the appropriate level for tests and full reporting of outcomes |
| ☒ | ☐ | Estimates of effect sizes (e.g. Cohen's *d*, Pearson's *r*), indicating how they were calculated |

*Our web collection on statistics for biologists contains articles on many of the points above.*

## Software and code

Policy information about availability of computer code

| | |
|---|---|
| Data collection | Reference genome<br>- Zostera marina v3.1 NCBI BioProject PRJNA701932<br><br>Downloading SRA data:<br>fasterq-dump (sratoolkit.2.10.8-centos_linux64)<br><br>novel sequencing data:<br>Illumina HiSeq4000 and NovaSeq6000 genetic analyzers and proprietary data collection software |
| Data analysis | Modeling and simulations<br>- package for running clonal organism simulations at https://github.com/jessierenton/SomaticEvolution.jl<br>- custom-made scripts and simulation data, including the implemented Gillespie-algorithm, at https://github.com/jessierenton/somatic-genetic-clock<br><br>Analyzing histological images via confocal microscopy of the shoot apical meristem<br>- converting original confocal z-stack images (LIF) to TIFF files using open source (Fiji, https://fiji.sc).<br>- processing images, open source MorphoGraphX (MGX) v.2.0.1  (https://morphographx.org/software/)<br><br>Quality check of the raw Next-Generation Sequencing data<br>- FastQC v0.11.7 (https://www.bioinformatics.babraham.ac.uk/projects/fastqc/). |

Filtering of the raw data
- BBDuk (https://jgi.doe.gov/data-and-tools/bbtools/bb-tools-user-guide/bbduk-guide/).

Mapping of short reads against reference genome
- sequence reads were mapped against the chromosome-level reference genome of Zostera marina V3.1 using BWA MEM  (Burrows-Wheeler Alignment Tool v0.1.17).
- alignments were converted to BAM format and sorted using Samtools v1.11
- MarkDuplicates module in GATK4 v4.1.1.0 was used to remove duplicated reads (repository for GATK4 package at https://github.com/broadinstitute/gatk)
- filtering of the bam files using Samtools v1.11

Joint-calling of single nucleotide polymorphism for the Estonian clones
- HaplotypeCaller (GATK4 v4.1.1.0) was used to generate a GVCF format file for each sample, GVCF files were combined by CombineGVCFs (GATK4 v4.1.1.0).
- GenotypeGVCFs (GATK4 v4.1.1.0) was used to call genetic variants.
- VariantsToTable (GATK4 v4.1.1.0) was used to extract INFO annotations.
- quality filtering: marking by VariantFiltration (GATK4 v4.1.1.0) accd. to the criteria MQ < 40.0; FS > 60.0; QD < 10.0; MQRandSum > 2.5 or MQRandSum < -2.5; ReadPosRandSum < -2.5; ReadPosRandSum > 2.5; SOR > 3.0; DP > 1380.04 (2 * average DP), and those SNPs were excluded by SelectVariants (GATK4 v4.1.1.0).
- clone assignment (i.e. clonemates) based on shared heterozygosity (custom-made script at https://github.com/leiyu37/Detecting-clonemates.git).

Calling of somatic genetic variation (SNPs -  single nucleotide polymorphisms)
- Mutect2 (GATK4 v4.1.1.0)
- Strelka2 (strelka-2.9.2.centos6_x86_64)

Mutational Spectra analysis
-germline: population-wise SNPs were extracted from 11705 core SNPs from Yu et al. Nature Plants 2023
--somatic SNPs were extracted from the 4 oldest genets detected in this data set
-mutational spectra were computed using the R-package  Mutational.Patterns (no version, accessed Jan 2024)

Calculating the variable VRF50(X1, X2) as proxy for fixed somatic genetic variation
- custom-made scripts at https://github.com/leiyu37/SomaticGeneticClock.git

creating maps (Fig.4; Supplementary Fig. 15):
https://www.qgis.org/en/site/

For manuscripts utilizing custom algorithms or software that are central to the research but not yet described in published literature, software must be made available to editors and reviewers. We strongly encourage code deposition in a community repository (e.g. GitHub). See the Nature Portfolio guidelines for submitting code & software for further information.

# Data

Policy information about availability of data

All manuscripts must include a data availability statement. This statement should provide the following information, where applicable:

- Accession codes, unique identifiers, or web links for publicly available datasets
- A description of any restrictions on data availability
- For clinical datasets or third party data, please ensure that the statement adheres to our policy

Custom-made scripts can be found at:
https://github.com/jessierenton/somatic-genetic-clock (simulation data)
https://github.com/jessierenton/SomaticEvolution.jl (analytical & population genetic calculations)
https://github.com/leiyu37/SomaticGeneticClock.git (bioinformatics)

Estonian eelgrass (Zostera marina) genets (=clones) (field sites KYD, SOE, KOI): BioProject no. PRJNA1025927
SRR26321797-SRR26321800
SRR26321805-SRR26321810

4-yr calibration clones: BioProject no. PRJNA1025927
SRR26321801-SRR26321804, SRR26321811, SRR26321812

17-yr calibration clones from California, Bodega Bay:
BioProject no. PRJNA806459
SRA accession nos. SRR18000159–SRR18000170.

Finnish clone (Ängsö):
BioProject no. PRJNA557092
SRA accession nos. SRR9879327- SRR9879353.

Eelgrass (Zostera marina) genets (=clones) sampled in global population genomics dataset (overview also given in Supplementary Data 1):
Bodega Bay, BB04, SRP193551
Bodega Bay, BB05, SRP193555
Bodega Bay, BB09, SRP193562
Bodega Bay, BB10, SRP193563

```
Japan South, JS03, SRP194687
Japan South, JS04, SRP193493
Northern Norway, NN02, SRP193666
Northern Norway,NN06, SRP193673
Northern Norway,NN07, SRP193674
Northern Norway,NN09, SRP194699
Northern Norway,NN10, SRP193677
Northern Norway,NN05, SRP193672
Northern Norway,NN08, SRP193675
Portugal, PO02, SRP193709
Portugal, PO05, SRP193715
Portugal, PO07, SRP194708
Portugal, PO08, SRP194712
Portugal, PO10, SRP194711
Portugal, PO11, SRP194715
Portugal, PO12, SRP194717
Portugal, PO03, SRP193716
Portugal, PO04, SRP193714
Portugal, PO06, SRP194707
Portugal, PO09, SRP194713
San Diego, SD04, SRP194696
San Diego, SD11, SRP193569
San Diego, SD06, SRP227665
San Diego, SD09, SRP193567
Washington State, WN04, SRP193698
Washington State, WN09, SRP227669
Washington State, WN06, SRP193703
Washington State, WN10, SRP227670
```

## Human research participants

Policy information about [studies involving human research participants and Sex and Gender in Research.](studies involving human research participants and Sex and Gender in Research.)

| | |
|---|---|
| Reporting on sex and gender | na |
| Population characteristics | na |
| Recruitment | na |
| Ethics oversight | na |

Note that full information on the approval of the study protocol must also be provided in the manuscript.

# Field-specific reporting

Please select the one below that is the best fit for your research. If you are not sure, read the appropriate sections before making your selection.

☐ Life sciences      ☐ Behavioural & social sciences      ☒ Ecological, evolutionary & environmental sciences

For a reference copy of the document with all sections, see [nature.com/documents/nr-reporting-summary-flat.pdf](nature.com/documents/nr-reporting-summary-flat.pdf)

# Ecological, evolutionary & environmental sciences study design

All studies must disclose on these points even when the disclosure is negative.

| | |
|---|---|
| Study description | The study combines agent based modeling on a hypothetical, generic clonal species, and empirical data of mixed origin on eelgrass (Zsotera marina) genets. In addition, confocal microscopy provided evidence for key parameters of growing eelgrass genets such as the stem cell population size, the founder population size and the ratio of symmetric vs. asymmetric cell divisions.<br>Data origin for empirical eelgrass data: Dataset of the 17-yr-old clones and of a global population genomic colelction of sites were from previous studies in which clone mates (i.e. ramets of the same genet), however, have not been analyzed . The dataset of the 4-yr-old calibration genets and the dataset of the Estonian clones were newly sequenced in this study. |
| Research sample | A research sample is a leaf shoot (or ramet) of the seagrass Zostera marina (=eelgrass). |
| Sampling strategy | Empirical data only: Samples were collected by snorkling or diving. Four-yr-old and 17-yr-old samples were collected from genets originally sampled at nearby locations (Kiel Bight, Germany, and Bodega Bay, California, USA; respectively), and cultured in the lab in large tanks (>500L)  under flow through of ambient seawater, experiencing outside light conditions, and rooted in ambient sediment. |

| Data collection | Collectors are mentioned in the section on sampling permits. Sample extraction was performed at GEOMAR Kiel (Diana Gill and Lei Yu). DNA samples were sent to BGI Genomics (Hong Kong) for Illumina sequencing. |
|---|---|
| Timing and spatial scale | Sampling for the previous population genomics project was conducted between May 2016 and August 2017. Estonian samples were collected in August 2021. At a given site, a population was defined as continuous eelgrass meadow of at least 50 m across (parallel to shore). Samples for the 4-yr-old clones were collected from the lab In 2022. |
| Data exclusions | SNPs not passing the filtering criteria were excluded. |
| Reproducibility | SNP calling: two independent SNP calling approches were used (STRELKA2, Mutect2)<br>calibration of the somatic genetic clock:  three (4-yr) and two (17-yr) old cultivated eelgrass genets (=clones) were used to obtain a calibration curve to age eelgrass genets (=clones) at other sites<br><br>identification of key covariates: an agent based model was used to examine the effects of branching rate (thus asexual generation time), number of founder cells, stem cell population size and the ratio asymmetric vs symmetric cell division. Within the parameter space of the study species eelgrass, a significant deviation of the somatic genetic clock from linearity is unlikely |
| Randomization | no randomization was required as the study question addressed identical clone mates |
| Blinding | no blinding was required |

Did the study involve field work?  ☒ Yes  ☐ No

## Field work, collection and transport

| Field conditions | As our study builds upon genome polymorphism and differentiation that was emerging over hundreds to thousands of years, no environmental data were collected at the time of sampling |
|---|---|
| Location | All 20 sampling locations were geo-referenced, coordinates are listed in Supplementary Data 1 and below |
| Access & import/export | For all sites, sampling permits have been obtained by the relevant national or regional authorities where required. An e-mail string can be provided upon request between the local collaborators and the respective national authorities (NFP) with respect to an obligation or waiver of CBD or general sampling permit.<br><br>Populations with presence of genets = clones with >= 2 rsmets sampled:<br>- Japan South /J S /Pos 34.298N 132.916E. Sampling: collecting permit to Dr. Masakazu Hori, CBD-"Nagoya": see above<br>-Bodega Bay, USA / BB /Pos 38.320N 123.055W. Sampling: permit to Dr. John S Stachowicz through Dept Fish Wildlife CA. CBD-"Nagoya": non-signatory<br>-San Diego Bay, USA / SD / Pos 32.714N 117.225W. Sampling: permit to Dr. Kevin A Hovel through Dept Fish Wildlife CA. CBD-"Nagoya":  non-signatory<br>-Røvika, Northern Norway / NN / Pos  67.268N 15.257E. Sampling: no permit required. CBD-"Nagoya": waiver<br>-Port Dinllaen, Wales, UK / WN/ 52.991N 4.450W. Sampling: waiver to Dr. Richard Unsworth by authorities as amount negligible. CBD-"Nagoya":  waiver /collection before 1 July 2017<br>-Ria Formosa, Portugal /PO / 37.040N 7.910W saapling: no collection permit required. CBD-"Nagoya": collection before 1 July 2017<br>- Baltic Sea /Estonia, site Kuedema/KYD/pos 58.5331N 22.2380E: sampling permit through Prof. Jonne Kotta, Univ Tartu, CBD-"Nagoya": waiver<br>- Baltic Sea /Estonia, site Soela Strait /SOE/ Pos 58.6420N 22.6036E: sampling permit through Prof. Jonne Kotta, Univ Tartu, CBD-"Nagoya": waiver<br>- Baltic Sea /Estonia, site Koinastu /KOI /Pos 58.6184N 22.9928E: sampling permit through Prof. Jonne Kotta, Univ Tartu, CBD-"Nagoya": waiver<br><br>Populations without clones (not further analyzed in this ms):<br>Specific information is listed below for each site, from West to East:<br>- Japan North / JN /Pos 43.021N 144.903E. Sampling: collecting permit to Dr. Massa Nakaoka (in Japanese). CBD-"Nagoya":  collection in August 2017 before implementation of CBD access regulation in Japan<br>- Alaska Safety Lagoon, USA /ASL / Pos 64.485N 164.762W. Sampling: no collecting permit required, waiver by U. S. Fish and Wildlife Service to Dr. David Ward & Dr. Sandra Talbot, CBD: non-signatory<br>-Alaska- Izembek Lagoon, USA /ALI / Pos 55.329N 162.821W. Sampling: no collecting permit required, waiver by U. S. Fish and Wildlife Service to Dr. David Ward & Dr. Sandra Talbot. CBD-"Nagoya": non-signatory<br>-Willapa Bay, Washington State, USA / WAS / Pos 46.474N 124.028W. Sampling: permit to Dr. Jennifer Ruesink through Wash Dept Natural Res. CBD-"Nagoya": non-signatory<br>-Quebec, Canada / QU / Pos 49.112N 68.176W. Sampling: permit to Dr. Mathieu Cusson through Fisheries and Oceans Canada. CBD-"Nagoya": non-signatory<br>-Massachusetts, USA / MA/  Pos 42.420N 70.915W. Sampling: permit to Dr. Randall Hughes through Massachusetts Division of Marine Fisheries. CBD-"Nagoya": non-signatory<br>-North Carolina, USA / NC / Pos 34.692N 76.623W. Sampling: permit to Dr. Joel Fodrie through North Carolina Division of Marine Fisheries. CBD-"Nagoya": non-signatory<br>-Torserød, West Coast of Sweden / SW / 58.313N 11.549E. Sampling: no permit required, waiver by Administrative County Board of Västra Götalands to Dr. Per-Olav Moksnes. CBD-"Nagoya": waiver<br>-Thau Lagoon, France / FR/ 43.447N 3.662E sampling: no collection permit required, waiver to Dr. Francesca Rossi. CBD-"Nagoya": |

| | waiver /collection before 1 July 2017 |
|---|---|
| | -Adriatic Sea, Croatia /CZ /Pos 44.212N 15.491E. sampling: no collection permit required, waiver to Dr. Stewart Schulz & Dr. Claudia Kruschel. CBD-"Nagoya": non-signatory |
| Disturbance | At each site, in an area of several 1000 m2, some leaf shoots of eelgrass were collected, representing <0.001% of all plants of the respective meadow. This level of disturbance is negligible compared to, for example, natural physical disturbance by storms or herbivory |

# Reporting for specific materials, systems and methods

We require information from authors about some types of materials, experimental systems and methods used in many studies. Here, indicate whether each material, system or method listed is relevant to your study. If you are not sure if a list item applies to your research, read the appropriate section before selecting a response.

## Materials & experimental systems

| n/a | Involved in the study |
|---|---|
| ☒ ☐ | Antibodies |
| ☒ ☐ | Eukaryotic cell lines |
| ☒ ☐ | Palaeontology and archaeology |
| ☒ ☐ | Animals and other organisms |
| ☒ ☐ | Clinical data |
| ☒ ☐ | Dual use research of concern |

## Methods

| n/a | Involved in the study |
|---|---|
| ☒ ☐ | ChIP-seq |
| ☒ ☐ | Flow cytometry |
| ☒ ☐ | MRI-based neuroimaging |

