## [Peer Review File · Nature Ecology & Evolution]

Peer Review Information

Journal: Nature Ecology & Evolution

Manuscript Title: A somatic genetic clock for clonal species

Corresponding author name(s): Benjamin Werner, Thorsten B. H. Reusch

Editorial Notes:

Reviewer Comments & Decisions:

Decision Letter, initial version:

9th January 2024

Dear Thorsten,

Your manuscript entitled "A somatic genetic clock for clonal species" has now been seen by four reviewers, whose comments are attached. The reviewers have raised a number of concerns which will need to be addressed before we can offer publication in Nature Ecology & Evolution. We will therefore need to see your responses to the criticisms raised and to some editorial concerns, along with a revised manuscript, before we can reach a final decision regarding publication.

We therefore invite you to revise your manuscript taking into account all reviewer and editor comments. Please highlight all changes in the manuscript text file in Microsoft Word format.

* If you have not done so already please begin to revise your manuscript so that it conforms to our Article format instructions at <http://www.nature.com/natecolevol/info/final-submission>. Refer also to any guidelines provided in this letter.

[REDACTED]

2Note: This URL links to your confidential home page and associated information about manuscripts you may have submitted, or that you are reviewing for us. If you wish to forward this email to co-authors, please delete the link to your homepage.

Nature Ecology & Evolution is committed to improving transparency in authorship. As part of our efforts in this direction, we are now requesting that all authors identified as 'corresponding author' on published papers create and link their Open Researcher and Contributor Identifier (ORCID) with their account on the Manuscript Tracking System (MTS), prior to acceptance. ORCID helps the scientific community achieve unambiguous attribution of all scholarly contributions. You can create and link your ORCID from the home page of the MTS by clicking on 'Modify my Springer Nature account'. For more information please visit www.springernature.com/orcid.

[REDACTED]

Reviewer expertise:

Reviewer #1: somatic mutations in plants

Reviewer #2: somatic mutation in tree of life

Reviewer #3: algal clones

Reviewer #4: somatic mutation in cancer

Reviewers' comments:

Reviewer #1 (Remarks to the Author):

Estimating age in clonal species is extremely challenging as it can be confounded by numerous factors. A somatic genetic molecular clock is therefore considered to be much harder to achieve when compared to the conventional genetic molecular clock. In this paper, the authors demonstrated the somatic genetic clock for clonal species is indeed achievable using a WGS approach aided by sophisticated modeling. They applied their methods to eelgrass, a promising model for clonal species, and demonstrated the power of such a somatic genetic clock. Their approach and modeling are

2promising and solve problems in estimating important life-history traits such as age in clonal species. They also provide more details about the SAM structure and development details of eelgrass. All these results are valuable and contributes to the understanding of the community.

I'm not expert in mathematical modeling so I will leave all these formulas to other experts. My questions are mainly about the generality of current somatic clock model when considering real development patterns in species like plants.

My first concern is that whether the proposed model is generic enough for most clonal species, especially for clonal plants? The authors found that the accumulation of neutral fixed SoGV is linear with time, providing the basis for the somatic genetic clock. Although they considered a range of scenarios with different types of stem cell division, different founder cell sizes, and varying rates and mechanisms for forming new modules, their models seem to always assume that the somatic cell population is forever "updating" (both during growth and homeostasis). This is possibly problematic for many plants with a clonal habitat but contain stem cells that remain quiescent for years (e.g., buds in dormancy), of these cells the somatic molecular clock is actually "turned-off" temporarily. However, these quiescent stem cells can replace the original population if they are later re-activated by environmental stimulating, which is a highly stochastic and possibly unpredictable process. I believe this is the frequent cases for land clonal plants as resources are generally very restricted so the chance to divide for difference stem cell niche varies greatly, but maybe for seagrass this is not the case.

Another confounding effect here which is not modeled is the competence between stem cell niches, e.g., between apical meristems and axillary meristems. The apical dominance leads to slower (or even stopped) proliferation rate in axillary meristematic stem cells compared to the apical ones, so the branching process is far from a purely random process. This again could cause the molecular clocks to run at different speeds in different ramets of a clonal plant.

Re-establishing of clonal population from an earlier quiescent or much slower divided stem cell niche could lead to re-setting of the somatic molecular clock. Hence, without consideration of the above development details, the current somatic model may be only limited to those clonal species which largely follow a manner as those somatic cells in cancer evolution. So I expect the authors to explain in more details about how these developmental details are considered in their models (if their models already cover these parts) or try to extend their models by incorporating these factors to allow their model better adapts to most clonal plants.

My second concern is about the modeling of layered structure. The authors examined the SAM structure of eelgrass and confirmed a two-layered structure as seen in most monocots. They further demonstrated that leaf tissues are derived exclusively from L1, so they could assume a one-layer case as leaf is used for bulk sequencing. One problem here is that for monocots, the L2 layer is long considered as the regeneration center, which means most founder stem cells for new ramets are likely derived from L2. How this would influence the modeling? Another problem is that eudicots generally have three layers, and L1 only contributes to the epidermis tissue of leaf. Is current model can also be used in this case? Overall, the SAM structure of eelgrass seems to be different from most plants, will the current model, which is heavily based on eelgrass, be too simple for other plants?

3Minor issues:

P2-L43: “the sexually produced offspring (genet)”, do the authors mean “asexually” here?

P5-L200: VRF50(Rx) estimation: The VRF value derived from NGS can be substantially influenced by stochastic events during library construction and sequencing. Will the accuracy of VRF50 estimation influence the calibration process?

P5-L204: The coverage is ~1000x for 4 years but only ~80x for 17 years. Why it varies so great?

Besides, what’s the recommended depth if we want to adopt current WGS approach to other species?

P7-L260: Did the authors compare the genetic based and epigenetic based clocks to see which one is possibly more robust to confounding factors? This would add more values to the current study.

Reviewer #2 (Remarks to the Author):

Overview and Significance:

This paper introduces a novel molecular clock based on the accumulation of fixed somatic genetic variation (SoGV) in clonal species, which is a significant contribution to understanding the age and longevity of these organisms. The authors employ a stochastic model to demonstrate the linearity of SoGV accumulation over time, considering factors like mitotic mutation rate, stem cell population size, and cell division types. The methodology appears robust, combining theoretical modelling with empirical data. The use of high-throughput sequencing and detailed genomic analysis strengthens the study. The application to eelgrass and the potential for broader use in other clonal species demonstrates the utility of the method and its potential relevance in broader studies. I have a few questions that pertain to the robustness of the method under particular scenarios and sampling conditions.

1. What is the rate of accumulation of fixed somatic mutations in eelgrass? The authors discuss the rate of fixed somatic genetic variants in eelgrass but it was not clear to me from the text what the actual rate was, i.e. how many somatic mutations are fixed per year (or other unit of time) in eelgrass? Perhaps I missed it but this seems like an important variable to make clear in the text. If it is very low it will have an impact on the confidence of the method to act as an accurate genetic clock over short timescales.

2. What if mutation rates vary between populations within a species? The authors use two populations of eelgrass to calibrate their somatic genetic clock. However it remains unclear to what extent somatic mutation rates are constant across individuals within a species. It is a reasonable assumption that they will be constant within species, however variation in somatic mutation rates, caused by genetic variation within a population or environmentally induced mutagenesis could alter mutation rates and impact the validity of the clock. While inherent variability in mutation rate is difficult to account for, it might be possible to detect environmentally induced mutations if the mutagen produces a particular mutational signature that can be distinguished from the baseline. Is this something the authors considered?

Related to this, did the authors look at the trinucleotide spectra of mutations they are detecting? A nice sanity check would be to compare this to the spectra of germline SNPs in the eelgrass population,

4as in theory they should be similar. Deviation from this could be reflective of sequencing artefacts (or mutations induced from environmental mutagens).

3. In their model the authors assume that mutations occur during each cell division. However recent results from human studies suggest that somatic mutations predominantly occur during ongoing cell maintenance and are not strongly correlated with cell division rate (e.g. Abascal et al. 2021). If this is the case then the model may not be dependent on cell division rates. This may in fact make it even more robust as a linear clock for calculating age. This is something the authors may want to mention in the text as I think it only strengthens the utility of their clock.

4. I assume that the choice of cells selected for bulk sequencing in each ramet could impact the results if different parts of the organisms are derived from different stem cell populations. Could the authors comment on this and the importance (or not) of the choice of bulk tissue for sequencing on the results? It might be useful to highlight this in the discussion for other research groups who may want to replicate the approach in different organisms.

5. I wonder if insights into the spatial patterns of development of eelgrass could be obtained using these data. Did the authors consider exploring if you could create a phylogenetic tree of the different ramets within an eelgrass to gain insights into how an eelgrass has grown over time, i.e. does it grow outwards in all directions from a central core? Clones that are closely related presumably share more fixed somatic variants and one might expect this to correlate with spatial proximity. Presumably the true growth pattern of eelgrass may be well characterised so mapping the somatic mutation rates onto these patterns could demonstrate the utility of this approach for learning about spatial growth patterns from other less well studied species using this method.

6. The authors state: "Fixed mutations within specific human tissues accumulate linearly with age 35,36." However the Abascal et al. 2021 reference does not show that fixed mutations accumulate with age but rather uses single molecule duplex sequencing to show that cell specific somatic variants accumulate with age, without reaching fixation in a bulk sample. Therefore this is not an appropriate reference for the point the authors are making.

In conclusion, the paper by Yu, Renton, et al. is a valuable contribution to the field, with potential applications in various biological disciplines. The study is well-conceived and executed, though some areas could benefit from further elaboration and clarification.

Reviewer #3 (Remarks to the Author):

This manuscript really caught my interest. A very nice story that combines advanced modelling with ambitious cell-morphology work to solve an issue that is essential to understand the demography and population dynamics of a clonal organism. The basic aim - to find a way to determine age of clones in a highly clonal species - is very interesting and truly generic, and this will be a useful tool for studies of wild clonal organisms. Based on an earlier finding of the authors - that neutral mutations are fixed

5by "somatic drift" in ramets formed from a few cells of a "mother" genet in clonal species - the authors now take the step to apply this knowledge to age determination. By modifying a model from cancer research the authors can describe the accumulation of somatic mutations in cell lines in a clonal organism. A crucial step is to identify the number of stem cells that give rise to each new clone, and the authors use advanced microscopy by which they study cell-division in the meristem in great detail. They also carefully investigate (by modelling, and using clones of known age) the preconditions for new mutations and are able to make some crucial conclusions from this, such as, how long time it takes until the mutation rate has stabilized and become a linear function of time, and hence independent of generation time under conditions of symmetric and asymmetric cell division. And with such a linear function, voilà - a molecular clock for clonal species! All in all, a very interesting and impressive study and I have only very few minor comments.

A general question I have (out of my ignorance) is how much the method rely on comparing ramets of similar age? Most invertebrates and plants have indeterminate growth but old individuals have slower growth (slower cell division) than young individuals, and so what will happen if one uses an old ramet and a young ramet, instead of comparing two ramets of the same life-stage. (For example, an old tree and a young vegetative shoot of the same genet.)

Moreover, how important is it that the calibration is done for the target species? Or do the authors consider it possible to use calibration from a related clonal species?

Detailed comments:

Line 28 - "macroalgae" instead of "algae" (excluding microalgae)

Line 70 - It would be useful to the reader if "symmetric" vs "asymmetric" cell division could be explored a bit more from early in the text (as it impacts speed of mutation rate quite a bit). How common are the different forms? And how common is it that they appear mixed, in the same individual (or in the same species)? It is mentioned later that both forms possibly occurred in the eelgrass species which is the target species of the study (line 126).

Line 117 - It is fairly obvious why number of stem cells and how many cells form a new module are important to know, but it is not clear to me why "stratification" is crucial. Can this be briefly explained here?

(On line 130 - there is a reference to stratification again, and here also it is not obvious why this is a potential problem, and how to escape this problem).

Line 186 - "geographic parthenogenesis" is not a hypothesis, I would rather call this a pattern. But there are many different hypotheses to explain this pattern.

Line 191 - there are two papers that explore the colonisation hypothesis much more directly than the Eckert 2002 paper here used as a reference. Please consult Rafajlovic et al. 2017 doi: 10.1111/jeb.13124 and Pereyra et al which are already in the list of references).

Line 192-194. This is an interesting conclusion, still the Finnish clone was estimated to be approx 1000 years old (which seem to be a good estimate) from size!?

Line 246 - "animal and fungal kingdom" excluding macroalgae ?!

Reviewer #4 (Remarks to the Author):

In this manuscript entitled "A somatic genetic clock for clonal species," Lei Yu and colleagues developed a statistical method that can be applied to clonal species to estimate their branching time. As mutations are thought to be acquired at a more-or-less constant rate over time, the numbers of sample-specific (clonal) mutations are molecular clocks, which can be used to estimate the differentiation time. Similar approaches have been applied to various biological questions, such as species differentiation, cancer clonal evolution, and early embryonic dynamics. The authors implement the equivalent idea to 'clonal reproduction'. The specific considerations for the issue are the unknown components in the actual process of module formation, such as branching/splitting, number of stem cells (N), number of founder stem cells in new modules (N_0), the rate of module formation (r), absolute mutation rate per cell per cell division (u), symmetric/asymmetric cell divisions, parts of which are well summarized in Figures 1 and 2. The topic is interesting, and their approach may have a massive application to various clonal species.

(1) The authors have generated a gorgeous statistical framework that can estimate the differentiation time of clonal species. The authors' deep consideration and their resultant equations are the most beautiful components of the work.

(2) However, the statistical model includes many different parameters that are usually unknown. For example, to apply their equations, we first need to know N , N_0 , and other parameters, which may need both experimental observations (as the author conducted (Figure S7-S9)) and its calibration using real-world data (the authors used a few samples with known ages). Although the equations are theoretically beautiful, their practical utility is uncertain. In addition, once some researchers know these parameters, statistical modeling will be relatively easy.

(3) Although the statistical model product is beautiful, it is not innovative, but something can be inferred from the logical inference.

(4) To prove the validity of their statistical model, the authors applied the models to eelgrass samples. To determine the N and N_e , potentially the most critical parameters, the authors experimentally observed the shoot apical meristem (SAM) of the eelgrass using confocal microscopy. Although the procedures were described in the methods and supplementary figures (Fig S7-S9), for authors, it is hard to understand how accurately the parameters are determined. Instead, the authors showed that their 'molecular clocks' were overall robust in the wide range of the N and N_e values (sentences #138-144). Suppose N and N_e are critical components for estimating the age. In that case, the experimental part should be shown more vigorously in the main manuscript with a main figure. Suppose accurate estimation of N and N_e is not critical. In that case, it may indicate that their sophisticated statistical

7modeling is unnecessary to estimate the branching time.

(4-1) Another issue is the authors need to explicitly show the sequencing data (the number of mutations) of many samples. For example, how many 'clonal mutations (or VRF50(Rx))' were there for each sample? What was the sequencing depth for each sample? How accurate were the mutation calls? Are there any structural events or large-scale gene conversions that may change the clonal mutation numbers?

(5) The main figures are beautiful, but many of them are 'cartoons', simulation data, or processed data that do not allow validation of their findings or 'critical reading' of the manuscript. Figures 1 and 2 are just illustrations of the 'models'. Figure 3 includes the simulation study rather than real-world observations. Figures 4c-4d are 'processed' data.

(6) To calibrate their model, they used sequencing data from a few samples (5 samples?) from two different time points (4 years and 17 years; Figure 4a). Although the authors showed the linear regression model, I think it is too crude, and the authors need more samples (observations).

(6-1). How many mutations were detected from each of the samples with known ages?

(7) The authors suggest that some samples from Estonia, Norway, and Finland are several hundred years old. How many mutations were detected in the samples? If the linear regression model (in Figure 4a) is applied to those samples, what are the estimated ages of the samples? What's the age difference between the linear regression model and the author's more sophisticated statistical model?

(8) How does the absolute mutation rate impact the age estimation? What were the mutational signatures (patterns) of the genomic mutations?

(9) minor point; Fig 1c should be updated. The current version seems that the mutation M3 is subclonal to the mutation M2, and the M4 is subclonal to M3.

*****END*****

Author Rebuttal to Initial comments

Yu et al. "A somatic genetic clock for clonal species" - Response-to-reviewer comments

8- our responses in blue -

Reviewer #1 (Remarks to the Author):

Estimating age in clonal species is extremely challenging as it can be confounded by numerous factors. A somatic genetic molecular clock is therefore considered to be much harder to achieve when compared to the conventional genetic molecular clock. In this paper, the authors demonstrated the somatic genetic clock for clonal species is indeed achievable using a WGS approach aided by sophisticated modeling. They applied their methods to eelgrass, a promising model for clonal species, and demonstrated the power of such a somatic genetic clock. Their approach and modeling are promising and solve problems in estimating important life-history traits such as age in clonal species. They also provide more details about the SAM structure and development details of eelgrass. All these results are valuable and contributes to the understanding of the community.

I'm not expert in mathematical modeling so I will leave all these formulas to other experts. My questions are mainly about the generality of current somatic clock model when considering real development patterns in species like plants.

My first concern is that whether the proposed model is generic enough for most clonal species, especially for clonal plants? The authors found that the accumulation of neutral fixed SoGV is linear with time, providing the basis for the somatic genetic clock. Although they considered a range of scenarios with different types of stem cell division, different founder cell sizes, and varying rates and mechanisms for forming new modules, their models seem to always assume that the somatic cell population is forever “updating” (both during growth and homeostasis). This is possibly problematic for many plants with a clonal habitat but contain stem cells that remain quiescent for years (e.g., buds in dormancy), of these cells the somatic molecular clock is actually “turned-off” temporarily. However, these quiescent stem cells can replace the original population if they are later re-activated by environmental stimulating, which is a highly stochastic and possibly unpredictable process. I believe this is the frequent cases for land clonal plants as resources are generally very restricted so the chance to divide for difference stem cell niche varies greatly, but maybe for seagrass this is not the case.

It is not necessary to assume that the somatic cell population is forever “updating”. The development history of a somatic cell population is actually a sequence of “turned-on” (when it is activated) and “turned-off” (when it is quiescent). The switch between “turned-on” and “turned-off” is of course determined by both stochastic and selective forces. Under the stochastic scenario, the clock still holds, and this process only slows down the clock to some extent. Any empirical calibration would consider the average cell depth (no of divisions) among module formation events. See more detailed comprehensive

9response below, and also an added section 1.4. to the Supplementary Notes, as well as a new Supplementary Figure 7.

Another confounding effect here which is not modeled is the competence between stem cell niches, e.g., between apical meristems and axillary meristems. The apical dominance leads to slower (or even stopped) proliferation rate in axillary meristematic stem cells compared to the apical ones, so the branching process is far from a purely random process. This again could cause the molecular clocks to run at different speeds in different ramets of a clonal plant.

Re-establishing of clonal population from an earlier quiescent or much slower divided stem cell niche could lead to re-setting of the somatic molecular clock. Hence, without consideration of the above development details, the current somatic model may be only limited to those clonal species which largely follow a manner as those somatic cells in cancer evolution. So I expect the authors to explain in more details about how these developmental details are considered in their models (if their models already cover these parts) or try to extend their models by incorporating these factors to allow their model better adapts to most clonal plants.

As the reviewer has pointed out developmental details are variable across species. In many cases, however, these details can be neglected when we consider their effect on various model parameters within our framework. For example, quiescence of stem cells during axillary meristem or bud dormancy will reduce the effective rate of stem cell divisions implemented in our model as the parameter (b). To estimate the rate of mutation fixation, we assumed a constant (b) value of approximately 122 divisions per year. Reducing this parameter (to account for meristem/bud dormancy) would have two effects: first it would proportionally increase conditional fixation times for mutation fixation by stem cell turnover through symmetric divisions (see Supplementary Note 1.3.1, Moran process). For example, for *Zostera*, we predicted relatively rapid fixation times (up to 0.1 year) through symmetric stem cells divisions. If we decrease the rate of stem cell divisions to approximately 60 or even 20 divisions per year, mutation fixation would occur within a year (i.e. up to 0.2 and 0.6 year, respectively). Thus, although potential quiescence of stem cells is expected to prolong the fixation time as well the lag phase before reaching linearity, mutations can ultimately be fixed within a relatively short time frame. Even in the limiting case where $b > 0$, fixation still occurs by formation of new modules, as is the case for purely asymmetric division, and thus, there is an upper bound on how far fixation times can be reduced. The second effect of reducing (b) is to reduce the effective mutation rate per year, and thus the rate of accumulation of fixed mutations. This is because we have assumed that mutations occur at cell division. As this rate is measured experimentally, and not inferred from the models, this does not present an issue in applying our method.

It should be noted that the shoot apical meristem in *Zostera marina* represents an extreme case with an unusual high number of stem cells (N) compared to other plants. Typically, only about 3-4 stem

10cells are reported in angiosperms, gymnosperms, and lycopods (reviewed in Burian 2021, cited). Such a low N would promote rapid mutation fixation due to homeostatic stem cell turnover through symmetric divisions, even if the rate of stem cell divisions were reduced due to temporary quiescence.

If quiescent modules do not form new modules, this would lead to a lower effective rate of module formation (r). Similarly, the phenomenon of apical dominance, would affect the rate of module formation when averaged over the population. Generally, a lower rate of module formation increases the time for mutation fixation, while a higher rate of module formation leads to faster fixation of mutations. For asymmetric cell division this increase is inversely proportional to (r) (see Supplementary Note 1.3.1, Eqs. 1.31-1.32). However, for symmetric cell division, particularly if the number of stem cells (N) is small, the effect is much smaller, because fixation primarily occurs by stem cell turnover which is independent of (r). Then, following the reviewer comments, the question arises how much difference is expected in different ramets of the same clone with respect to stem cell division and module formation rates.

Apical dominance (along with related axillary meristem or bud dormancy) is determined by endogenous physiological, genetic, and developmental factors, such as the timing of vegetative and reproduction phases, phytohormone levels (auxin, cytokinin, strigolactans) as well as by exogenous environmental factors like temperature, light, mechanical wounding by herbivores. While some factors (such as herbivory) may act randomly on individual ramets, leading to temporally higher cell division or/and branching rate, they will not change the long-term stem cell or branching dynamics. Other factors, such as developmental phase, hormone levels, or any seasonal changes, are expected to exert a similar effect on the entire ramet population. Currently, we do not have any detailed data on rates and branching mechanisms in seagrass populations, nevertheless, we will explore this aspect in our future studies. Based on the analysis of a range of parameter values in our model (Fig. 3; Suppl Fig. 2-6; 12-13), the genetic clock can accommodate the potential differences that appear in the ramet population.

Based on the considerations above, we have added a new Supplementary Figure 7 and the following subsection 1.4. to Supplementary Note 1.

"1.4 Variation in developmental details and mutational processes

Thus far we have considered a relatively simple model of a clonal organism, in which cell division and module formation rate are constant in time and across the population, and mutations occur only at cell division. Of course, the details of developmental processes will vary across species. For example, stem cells may not continuously divide and modules may enter periods of quiescence. Furthermore, in many plants, apical dominance will result in varying module formation rates across the organism. Although it is beyond the scope of this work to consider the vast array of developmental regimes across clonal species, we have considered two examples in order to illustrate the robustness of our results to variation in

biological detail. These are (i) *stochastic quiescence*: homeostatic modules move in (and out) of a quiescent state with a fixed rate; and (ii) *seasonal quiescence*, all modules become quiescent during a winter period. We also consider the possibility that mutations may occur during cell lifetime, as well as at cell division. To this end we introduce a time-dependent mutation rate ξ in addition to the per cell mutation rate μ . Supplementary Fig. 7 shows the time-evolution of fixed SoGV under these different regimes, as well as the estimated accumulation rate of fixed mutations, in a population of modules that divide by asymmetric cell division. The lag-phase before linearity is reduced for both quiescence regimes (Supplementary Fig. 7a-c), indicating that the average rate of module formation across the population is lowered. However, linearity is still reached in all cases (Supplementary Fig. 7d-f). Furthermore, if mutations occur solely at cell division ($\mu=0.1$, $\xi=0.0$) or at cell division and during cell lifetime ($\mu=0.06$, $\xi=6$), the rate of accumulation of fixed mutations is reduced in the quiescence regimes compared to no quiescence, indicating that the average rate of cell divisions is reduced (Supplementary Fig. 7 d-f). When mutations are solely time-dependent ($\mu=0.0$, $\xi=12$), the rate of accumulation of fixed SoGV is exactly equal to ξ and does not depend on the quiescence regime, because there is no longer any dependence on the cell division rate. These results suggest that the somatic genetic clock is robust to stochastic or seasonal quiescence and that we can adapt our theoretical results simply by estimating r and b as an average over the population and over time.”

The following addition was to the manuscript (L135):

"Finally, we also considered additional complications with respect to the developmental mode of the clonal organisms. Under (i) *stochastic quiescence*, homeostatic modules move in (and out) of a quiescent state with a fixed rate; while under (ii) *seasonal quiescence*, all modules become quiescent during a winter period (Supplementary Note 1). We also consider the possibility that mutations may occur during cell lifetime, as well as at cell division. To this end we introduce a time-dependent mutation rate ξ in addition to the per cell mutation rate μ (Supplementary Note 1, Supplementary Fig. 7). The lag-phase before linearity is increased for both quiescence regimes (Supplementary Fig. 7a-c), indicating that the average rate of module formation across the population is lowered. However, linearity is still reached in all cases."

My second concern is about the modeling of layered structure. The authors examined the SAM structure of eelgrass and confirmed a two-layered structure as seen in most monocots. They further demonstrated that leaf tissues are derived exclusively from L1, so they could assume a one-layer case as leaf is used for bulk sequencing. One problem here is that for monocots, the L2 layer is long considered as the regeneration center, which means most founder stem cells for new ramets are likely derived from L2. How this would influence the modeling? Another problem is that eudicots generally have three layers, and L1 only contributes to the epidermis tissue of leaf. Is current model can also be used in this case?

12Overall, the SAM structure of eelgrass seems to be different from most plants, will the current model, which is heavily based on eelgrass, be too simple for other plants?

Generally, in shoot apical meristems with a layered structure, each cell layer has its own population of stem cells which give rise to separate cell lineages, where cells do not mix between layers. Thus, the propagation of mutations through cell divisions is also restricted to particular layers and all derived tissues. Indeed, two manuscripts posted recently on BioRxiv (Amundson et al. 2023 ‘Differential mutation accumulation in plant meristematic layers; Goel et al. 2024 ‘The majority of somatic mutations in fruit trees are layer-specific’, both now cited) show that, at least in the analyzed dicot species (potato, apricot), detected mutations were layer-specific. Thus, given that each meristem cell layer behaves autonomously, it can be treated as a separate entity.

Hence, for simplicity, and also based on previous evidence (cf. Supplementary Figure 11) we only modelled a one-layer case (L1) as the majority of studies focus on cellular events in this outermost layer, which is the most accessible for observations. Consequently, knowledge about stem cells in underlying cell layers is very limited. However, assuming that cell growth and division dynamics are similar in different meristem layers (Jackson et al. 2019, cited), our model can also be applied to any other cell layer.

To connect the model with the sequencing data, one has to characterize the mutations fixed in a specific layer. Thus, in this context, the choice of tissue or organ for genome sequencing is important. Our seagrass samples (‘leaf shoots’) contained several basal portions of leaves in addition to the shoot apex. We identified >7000 completely fixed mutations in the oldest and largest clone (Yu et al., 2020, cited). Also, a simpler fixation dynamic with a clear mode at $f = 0.5$ becomes apparent from the VRFs on all (fixed and mosaic) mutations (see Suppl Fig. 11). Given that cells from L1 layer generate most, if not all, leaf tissues (see Suppl Fig. 9), the presence of fixed mutations suggests that our bulk sampling represents one layer and is thus compatible with the model.

Regarding the origin of new ramets in monocots, we observed that the layered structure is maintained during the initiation of axillary meristems in *Zostera*. This means that stem cells at L1 and L2 of the apical meristem give rise to L1 and L2 cells of the axillary meristem, respectively. As the cell division plane is not restricted in the L2, this region may generate much more cells and tissues in the axillary meristem and in the stem of a new ramet compared to the L1. However, L1-derived cells would still dominate in leaf tissues. This suggests that in this respect, *Zostera* is similar to other monocots. In the case of dicots, indeed, the L1 only contributes to the epidermis in leaves, but a significant portion of inner cells in leaf blade can derive from L2 or/and L3 (Poethig 1989; Szymkowiak and Sussex 1996 cited in supplement). A manuscript posted recently in BioRxiv (Goel et al. 2024 ‘The majority of somatic mutations in fruit trees are layer-specific’, now cited) implies that in bulked leaf tissues from apricot trees,

the L2-derived mesophyll cells dominate. Thus, a dicot leaf may be a good representative for estimating the mutation rate at L2 or L3.

Changes in the manuscript were: main text, L151-152 – one sentence added on the similarity of meristem layers. Second, we modified the Suppl note 2.2. (before the last sentence)

"Because the cell division plane is not restricted in the L2, it has the capacity to produce a larger number of cells and tissues within the axillary meristem itself and in the future stem of a new ramet compared to the L1. Nonetheless, cells derived from the L1 would still dominate in leaf tissues."

We added to the Suppl note 2.3. "In contrast, in dicots characterized by two-layered tunica (L1 and L2) and corpus (L3) meristem structure, leaf tissues predominantly consist of cells derived from the L2 or/and L3 (Poethig 1989; Szymkowiak and Sussex 1996). However, there exist species-specific variations in the contribution of these cell layers in the ultimate structure of a leaf."

These refs were added:

Poethig, S. (1989). Genetic mosaics and cell lineage analysis in plants. *Trends in Genetics*, 5, 273-277.

Szymkowiak, E. J., & Sussex, I. M. (1996). What chimeras can tell us about plant development. *Annual review of plant biology*, 47(1), 351-376.

Minor issues:

P2-L43: "the sexually produced offspring (genet)", do the authors mean "asexually" here?

We here indeed mean sexually produced, as in originating from recombination happening when going through the zygote stage. This is the beginning of every genet in a clonal species

P5-L200: VRF50(Rx) estimation: The VRF value derived from NGS can be substantially influenced by stochastic events during library construction and sequencing. Will the accuracy of VRF50 estimation influence the calibration process?

Point well taken, of course any inaccuracy will influence VRF estimation and hence, the age determination. To examine the effects of coverage and sequencing platform, we have added two new Supplementary Tables 2, 3, and found little effects. Also in terms of the actual mapping and calling, we applied stringent criteria. VRF50(X1, X2) was calculated as the number of SNPs meeting the following criteria: 1) the coverage of X1 ≥ 12 ; 2) the coverage of X2 ≥ 23 ; 3) the variant read frequency of X1 ≤ 0.01 ; 4) the variant read frequency of X2 ≥ 0.50 . For the lower boundary: X1 has 12 reads, and no reads support the variant alleles, while X2 has 23 reads and 12 reads support the variant allele. This is quite a substantial difference. Although we could not 100% exclude the possibility of library and sequencing bias, the effects are supposed to be weak. Finally, we found only little effect of the mutation caller package (see new Supplementary Table 1).

Thus, while we exclude a bias of the somatic mutation calling, of course the variance will be affected both by the accuracy of every single, genet-wise differentiation value, as well by the no of genets of a given age that can be used for the calibration.

P5-L204: The coverage is $\sim 1000x$ for 4 years but only $\sim 80x$ for 17 years. Why it varies so great? Besides, what's the recommended depth if we want to adopt current WGS approach to other species?

The intention for the ultra-high coverage data set was to look at sub-fixation, mosaic mutations. We also subsampled the read coverage to a uniform $80x$ and found little difference in the absolute number of detected fixed SNPs (see new Supplementary Table 2). We hence present the most comprehensive data set in the main manuscript as the heterogeneous coverage seemed to matter little. Also note our site-specific coverage requirements that were similar across all samples (previous question).

P7-L260: Did the authors compare the genetic based and epigenetic based clocks to see which one is possibly more robust to confounding factors? This would add more values to the current study.

Epigenomic data are not yet available for our population data set, but surely this would be a very interesting comparative study to perform in the future.

Reviewer #2 (Remarks to the Author):

Overview and Significance:

This paper introduces a novel molecular clock based on the accumulation of fixed somatic genetic

variation (SoGV) in clonal species, which is a significant contribution to understanding the age and longevity of these organisms. The authors employ a stochastic model to demonstrate the linearity of SoGV accumulation over time, considering factors like mitotic mutation rate, stem cell population size, and cell division types. The methodology appears robust, combining theoretical modelling with empirical data. The use of high-throughput sequencing and detailed genomic analysis strengthens the study. The application to eelgrass and the potential for broader use in other clonal species demonstrates the utility of the method and its potential relevance in broader studies. I have a few questions that pertain to the robustness of the method under particular scenarios and sampling conditions.

1. What is the rate of accumulation of fixed somatic mutations in eelgrass? The authors discuss the rate of fixed somatic genetic variants in eelgrass but it was not clear to me from the text what the actual rate was, i.e. how many somatic mutations are fixed per year (or other unit of time) in eelgrass? Perhaps I missed it but this seems like an important variable to make clear in the text. If it is very low it will have an impact on the confidence of the method to act as an accurate genetic clock over short timescales.

We have added values of the estimated number of fixed mutations as a new column to the Supplementary Data1, named “N_fixed”, assuming that the distribution of VRFs in case of $f=0.5$ (i.e. heterozygosity) is symmetric. Along with the estimate of the yearly fixation rate of somatic variation among ramets, we have also added a sentence to the results that the rate of accumulation of fixed somatic mutations = 0.5044×2 based on an effective (average mappable) genome size of 209 Mbp, which translates to 4.6×10^{-9} /yr/bp. Interestingly, this estimate is rather close to the one for *Arabidopsis* which is at 7×10^{-9} /yr/bp.

This mutation estimate has now been added to the main ms.

2. What if mutation rates vary between populations within a species? The authors use two populations of eelgrass to calibrate their somatic genetic clock. However it remains unclear to what extent somatic mutation rates are constant across individuals within a species. It is a reasonable assumption that they will be constant within species, however variation in somatic mutation rates, caused by genetic variation within a population or environmentally induced mutagenesis could alter mutation rates and impact the validity of the clock.

We agree that different mutation rates among populations or species or clades are always a caveat in any molecular clock. As this is then a very generic statement, we have refrained from mentioning this in the text.

While inherent variability in mutation rate is difficult to account for, it might be possible to detect environmentally induced mutations if the mutagen produces a particular mutational signature that can be distinguished from the baseline. Is this something the authors considered?

Related to this, did the authors look at the trinucleotide spectra of mutations they are detecting? A nice sanity check would be to compare this to the spectra of germline SNPs in the eelgrass population, as in theory they should be similar.

Deviation from this could be reflective of sequencing artefacts (or mutations induced from environmental mutagens).

This is a good suggestion. Accordingly, we have compared the mutation spectra based on 1) the somatic SNPs of the four oldest clones, 2) the germline SNPs from a population genomic dataset recently published in Nature Plants (that also served as basis for the clone detection /ageing), now added as a new Supplementary Figure 16. Both mutational patterns show the well-known described predominance of transitions over transversions in plants. In other words, there are only subtle differences among germline and clonal SNPs, while the overall pattern is "plant-like" and does not raise any concerns on the validity of results.

Accordingly, we made a small addition to the main ms (L253) which reads "We also examined the mutational spectra of the 4 oldest genets detected here to six sexually segregating North Atlantic populations to identify possible differences between germline and asexually generated, mitotic mutations (Supplementary Figure 16). Both mutational patterns show the well-known predominance of transitions over transversions in plants {Ossowski, et al. Science 2010 now cited}. In particular, G:C->A:T transitions contribute (mean±1SD; n=6 and 4, respectively) 61±4 and 66±3% of all SNPs in asexual vs. sexual populations, respectively, regardless of their trinucleotide context (see Supplementary Figures 16a,b)."

3. In their model the authors assume that mutations occur during each cell division. However recent results from human studies suggest that somatic mutations predominantly occur during ongoing cell maintenance and are not strongly correlated with cell division rate (e.g. Abascal et al. 2021). If this is the case then the model may not be dependent on cell division rates. This may in fact make it even more robust as a linear clock for calculating age. This is something the authors may want to mention in the text as I think it only strengthens the utility of their clock.

This is a really good point. Although most of somatic mutations in plants were believed to be generated due to DNA replication errors, also other factors (like UV radiation or other mutagens) might contribute to higher mutation rate independently on cell divisions.

For example, recent study showed that mutation rate in tropical trees is not correlated with growth dynamics, which lead to the conclusion that mutations in plants can accumulate also with age (Satake et al. 2023, eLife, 'Somatic mutation rates scale with time not growth rate in long-lived tropical trees', now cited). Indeed, if mutations are predominantly time-dependent, rather than occurring at cell division, then the mutation rate is robust to changes in developmental detail (like module formation rate, cell division rate, periods of quiescence). We have added an additional Supplementary Figure 7 to highlight this that incorporates time-dependent and division-dependent mutational processes.

We have added the following passage to the discussion section of the manuscript (L328):

"Currently, we cannot distinguish mutations resulting from DNA replication errors during mitotic divisions from those occurring outside cell division. Indeed, recent studies suggest that somatic mutations can also accumulate with age in both plants and animals (Abascal et al. 2021; Satake et al. 2023). This indicates that, independently of cell division dynamics, other factors like UV radiation, transposons, or insufficient DNA repair systems could also increase the accumulation of mutations over time. A comparison of mutational spectra (Supplementary Figure 16a,b) does not suggest that the frequency of a type of transitions commonly associated with environmental stress in plants, such as UV (G:C -> A:T) is increased under long-term clonal growth. Even if this was the case in other species, it would rather enhance the validity of our somatic genetic clock, as it decouples somatic mutation accumulation from developmental processes."

4. I assume that the choice of cells selected for bulk sequencing in each ramet could impact the results if different parts of the organisms are derived from different stem cell populations. Could the authors comment on this and the importance (or not) of the choice of bulk tissue for sequencing on the results? It might be useful to highlight this in the discussion for other research groups who may want to replicate the approach in different organisms.

See also our detailed response to reviewer #1.

In our model, only the mutations fixed during the ramet formation matter. The seagrass case is simple, as each ramet only contains one SAM. For cases where one ramet has multiple SAMs, such as clonal trees, detecting SNPs based on one SAM of the ramet may include the mutations sub-fixed in this specific

SAM. For such cases, we would recommend to collect multiple samples from one ramet, and try to detect the shared ones characterizing one ramet as a whole.

If tissue derived from a particular layer can be identified (see e.g. in Goel et al. 2024 ‘The majority of somatic mutations in fruit trees are layer-specific’, BioRxiv) then the clock is easily applicable. Note that the layered SAM structure most likely only applies to angiosperm plants.

5. I wonder if insights into the spatial patterns of development of eelgrass could be obtained using these data. Did the authors consider exploring if you could create a phylogenetic tree of the different ramets within an eelgrass to gain insights into how an eelgrass has grown over time, i.e. does it grow outwards in all directions from a central core? Clones that are closely related presumably share more fixed somatic variants and one might expect this to correlate with spatial proximity. Presumably the true growth pattern of eelgrass may be well characterised so mapping the somatic mutation rates onto these patterns could demonstrate the utility of this approach for learning about spatial growth patterns from other less well studied species using this method.

This is an interesting point, yet our data do only permit limited insights into this question. The spatial pattern for the Estonia clone sampling can be found in supplement, here only targeted, molecular marker-identified ramets of a few genets were deep sequenced, not permitting a thorough spatial analysis (Supplementary Fig. 15). All other samples were not mapped. Only the large genet previously characterized in Yu et al. (Nat Ecol Evol 2020) permits an assessment of congruence among spatial position and phylogenetic patterns. As it turns out, the phylogeny does not fully match spatial proximity, most likely due to uprooting of vegetative fragments and their re-rooting someplace else in the area.

6. The authors state: “Fixed mutations within specific human tissues accumulate linearly with age 35,36.” However the Abascal et al. 2021 reference does not show that fixed mutations accumulate with age but rather uses single molecule duplex sequencing to show that cell specific somatic variants accumulate with age, without reaching fixation in a bulk sample. Therefore this is not an appropriate reference for the point the authors are making.

The referee is correct. Our formulation was a bit sloppy here. Indeed, mutations accumulate linearly with age in single stem cells and overall burden increases linearly with age. In most tissues, these mutations lead to local (space) fixation, examples are skin or colon, where tissue is organized into crypts. We modified the relevant section to "Somatic mutations accumulate linearly with age in human stem cells and

fixate at a constant rate locally in spatially constrained stem cell populations, e.g. in colon crypts or skin {Abascal, 2021 #4801; Blokzijl, 2016 #4800}".

In conclusion, the paper by Yu, Renton, et al. is a valuable contribution to the field, with potential applications in various biological disciplines. The study is well-conceived and executed, though some areas could benefit from further elaboration and clarification.

Reviewer #3 (Remarks to the Author):

This manuscript really caught my interest. A very nice story that combines advanced modelling with ambitious cell-morphology work to solve an issue that is essential to understand the demography and population dynamics of a clonal organism. The basic aim - to find a way to determine age of clones in a highly clonal species - is very interesting and truly generic, and this will be a useful tool for studies of wild clonal organisms. Based on an earlier finding of the authors - that neutral mutations are fixed by "somatic drift" in ramets formed from a few cells of a "mother" genet in clonal species - the authors now take the step to apply this knowledge to age determination. By modifying a model from cancer research the authors can describe the accumulation of somatic mutations in cell lines in a clonal organism. A crucial step is to identify the number of stem cells that give rise to each new clone, and the authors use advanced microscopy by which they study cell-division in the meristem in great detail. They also carefully investigate (by modelling, and using clones of known age) the preconditions for new mutations and are able to make some crucial conclusions from this, such as, how long time it takes until the mutation rate has stabilized and become a linear function of time, and hence independent of generation time under conditions of symmetric and asymmetric cell division. And with such a linear function, voilà - a molecular clock for clonal species!

All in all, a very interesting and impressive study and I have only very few minor comments.

A general question I have (out of my ignorance) is how much the method rely on comparing ramets of similar age? Most invertebrates and plants have indeterminate growth but old individuals have slower growth (slower cell division) than young individuals, and so what will happen if one uses an old ramet and a young ramet, instead of comparing two ramets of the same life-stage. (For example, an old tree and a young vegetative shoot of the same genet.)

Our method does not rely on comparing ramets of similar age. In our model, only the mutations fixed during the ramet formation matter. The seagrass case is simple, as each ramet only contains one SAM, and is also young. For clonal trees, the key would still be to detect the mutations fixed during the ramet formation. Therefore, we would recommend to collect multiple samples from one ramet, and try to detect

20the shared ones. Important would be that ramets of the same age are sampled, which in turn depends on a proper ramet definition. In the simple case eelgrass this is straightforward, as every leaf shoot bundle is defined as the ramet which is of the same age. By definition all ramets of the same clone have the same age (as they are descendants of the same zygote), so that growth slows down in only some is unlikely.

Moreover, how important is it that the calibration is done for the target species? Or do the authors consider it possible to use calibration from a related clonal species?

Probably, a proper calibration has to be done for each target species. Once more data are collected, one might generalize over certain clonal species with a shared developmental pattern. To do so, one would have to verify that cell division and stem cell population size are similar among different species. At the moment we do not know enough. In particular, it is unknown how exactly new modules are formed in clonal animals, say, corals. For the large group of clonally reproducing higher plants, despite the similarities in meristem organization, stem cell dynamics, specification of founder cells for axillary meristem, and branching mechanisms among closely related species, there may still exist species-specific variations, which could impact mutation rates. Thus, we would recommend calibrating for each target species, ideally growing under similar conditions, to account the potential influence of environmental factors.

Detailed comments:

Line 28 - "macroalgae" instead of "algae" (excluding microalgae)
has been changed accordingly

Line 70 - It would be useful to the reader if "symmetric" vs "asymmetric" cell division could be explored a bit more from early in the text (as it impacts speed of mutation rate quite a bit). How common is the different forms? And how common is it that they appear mixed, in the same individual (or in the same species)? It is mentioned later that both forms possibly occurred in the eelgrass species which is the target species of the study (line 126).

Both asymmetric and symmetric division modes occur in the population of plant stem cells independently of species, as they have been observed in different plant lineages, such as angiosperms (both dicots and

monocots), gymnosperms, and lycopods (Klekowski 1988; Gola and Jernstedt 2011; Zagorska-Marek and Turzanska 2000; Conway and Drinnan 2017). However, there are likely species-specific variations with the relative proportions of these division modes. For example, high stem cell turnover is predicted for *Selaginella* (Harrison et al. 2007), while stem cells are predicted to be relatively stable in privet (*Ligustrum*) shrub (Steward and Derman 1970). However, to directly estimate the frequency of asymmetric versus symmetric divisions, long-term time-lapse imaging is needed, which until now has been only applied to the model plant species such as *Arabidopsis*. We expect to develop similar technique for *Zostera* in our future studies to reveal dynamics of stem cells.

The following changes to the ms were made: in the main text, L91-92, a half sentence was introduced to explain how those two modes of stem cells divisions determine cell dynamics. Accordingly, we added the following section to the Suppl Note 2.4 (to the end of the 3rd paragraph): "Although both symmetric and asymmetric division modes characterize plant stem cell dynamics, species-specific differences likely exist in their occurrence frequency. For example, using indirect methods, frequent symmetric divisions leading to stem cell turnover are predicted for some ferns (Harrison et al. 2007), while asymmetric divisions likely dominate in angiosperms (Burian 2021). However, direct estimation of the frequency of these divisions requires long-term time-lapse imaging, a technique currently limited to the model plant species like *Arabidopsis* (Burian et al. 2016)." A novel reference was added to the supplement "Harrison CJ, Rezvani M, Langdale JA (2007) Growth from two transient apical initials in the meristem of *Selaginella kraussiana*. *Development* 134: 881-889 doi 10.1242/dev.001008"

Line 117 - It is fairly obvious why number of stems cells and how many cells form a new module are important to know, but it is not clear to me why "stratification" is crucial. Can this be briefly explained here? (On line 130 - there is a reference to stratification again, and here also it is not obvious why this is a potential problem, and how to escape this problem).

Stratification refers to independent cell subpopulations within an organism, derived from different sets of founder stem cells. Unless a tissue derived from one layer predominates (as is the case in eelgrass), under bulk sequencing this might cause mixing of cell populations with their specific mutations, causing variant read frequency patterns that make calling fixed SNPs impossible.

We have explained the importance of meristem stratification in the Supplementary Note 2.1, and also added a short paragraph to the discussion (L338): "The stem cell population dynamics during module formation is currently unknown for most clonal species other than angiosperms. The latter are complicated due to stem cell stratification into layers. However, even under these circumstances the somatic genetic clock can be applied when either the sampled tissue is dominated by one meristematic

layer (as is the case in eelgrass *Z. marina*), or when descendant tissues of a certain stem cell population can be clearly distinguished among the adult plant organs {Goel et al 2024 now cited}."

Line 186 - "geographic parthenogenesis" is not a hypothesis, I would rather call this a pattern. But there are many different hypotheses to explain this pattern.

has been changed accordingly

Line 191 - there are two papers that explore the colonisation hypothesis much more directly than the Eckert 2002 paper here used as a reference. Please consult Rafajlovic et al. 2017 doi: 10.1111/jeb.13124 and Pereyra et al which are already in the list of references).

has been changed accordingly

Line 192-194. This is an interesting conclusion, still the Finnish clone was estimated to be approx 1000 years old (which seem to be a good estimate) from size!?

Indeed, we had originally estimated the clone at approximately 750-1500 yrs, based on spatial extent divided by yearly lateral expansion of 20 cm / yr from a hypothetical centre (Yu et al. Nature Evol Ecol 2020). Indeed, the congruence among this method based on spatial extent and the genetic assessment is consistent with our molecular method. But we can envisage many other situations where this would not be the case, i.e. when only a small section of the clone has been samples, or when there were multiple cycles of growth and shrinking of a genet.

Line 246 - "animal and fungal kingdom" excluding macroalgae ?!

has been changed accordingly

Reviewer #4 (Remarks to the Author):

In this manuscript entitled "A somatic genetic clock for clonal species," Lei Yu and colleagues developed a statistical method that can be applied to clonal species to estimate their branching time. As mutations are thought to be acquired at a more-or-less constant rate over time, the numbers of sample-specific (clonal) mutations are molecular clocks, which can be used to estimate the differentiation time. Similar approaches have been applied to various biological questions, such as species differentiation, cancer

23clonal evolution, and early embryonic dynamics. The authors implement the equivalent idea to 'clonal reproduction'. The specific considerations for the issue are the unknown components in the actual process of module formation, such as branching/splitting, number of stem cells (N), number of founder stem cells in new modules (N_0), the rate of module formation (r), absolute mutation rate per cell per cell division (u), symmetric/asymmetric cell divisions, parts of which are well summarized in Figures 1 and 2. The topic is interesting, and their approach may have a massive application to various clonal species.

(1) The authors have generated a gorgeous statistical framework that can estimate the differentiation time of clonal species. The authors' deep consideration and their resultant equations are the most beautiful components of the work.

We thank the reviewer for the appreciation of our work

(2) However, the statistical model includes many different parameters that are usually unknown. For example, to apply their equations, we first need to know N , N_0 , and other parameters, which may need both experimental observations (as the author conducted (Figure S7-S9)) and its calibration using real-world data (the authors used a few samples with known ages). Although the equations are theoretically beautiful, their practical utility is uncertain. In addition, once some researchers know these parameters, statistical modeling will be relatively easy.

We show theoretically that the somatic genetic clock can work, and verify this using microscopy as well as VRF diagrams and plotting fixed mutations vs. known clone ages. The applicability of our approach is potentially universal for clonal species that regularly form modules, a very large group of animals, plants, macroalgae and fungi. The primary utility of the equations is that they allow for quick calculations of the time scale over which the somatic genetic clock is applicable for a particular species. This can be done with estimated parameters.

(3) Although the statistical model product is beautiful, it is not innovative, but something can be inferred from the logical inference.

We agree that our model is rooted in ideas of clonal dynamics and molecular clocks.

However, the model is a significant extension of molecular clocks used in simple clonal organisms like cancer populations. It introduces a two-level population (population formed of modules formed of cells) and the dynamics become more intricate. The mathematical results are an extension of the classical

Wright-Fisher model with the introduction of an additional sampling step, allowing us to represent the two-level population effectively, and crucially, to elucidate dependence on N_0 .

(4) To prove the validity of their statistical model, the authors applied the models to eelgrass samples.

We here wish to point out that we do not validate our model with seagrass, but rather apply the model to a species. We deduce from the model that given the boundary conditions laid out in the ms, in particular L107-145, the clock is valid and should be linear, except a short lag phase initially.

To determine the N and N_e , potentially the most critical parameters, the authors experimentally observed the shoot apical meristem (SAM) of the eelgrass using confocal microscopy. Although the procedures were described in the methods and supplementary figures (Fig S7-S9), for authors, it is hard to understand how accurately the parameters are determined. Instead, the authors showed that their 'molecular clocks' were overall robust in the wide range of the N and N_e values (Lxxx). Suppose N and N_e are critical components for estimating the age. In that case, the experimental part should be shown more vigorously in the main manuscript with a main figure. Suppose accurate estimation of N and N_e is not critical. In that case, it may indicate that their sophisticated statistical modeling is unnecessary to estimate the branching time.

We here combine the best of both worlds: we first introduce why such a somatic genetic clock is valid for a wide parameter set and life-history diversity of clonal species. In the second half, we highlight that in *Z. marina*, we most likely operate within a parameter space which is well within these limits.

We also note that the N and N_0 are most important for the lag time, not the linearity of clock in itself. To this end, we wanted to make sure that a calibration point at only 4 yrs is already useful in seagrass, which we found to apply.

(4-1) Another issue is the authors need to explicitly show the sequencing data (the number of mutations) of many samples. For example, how many 'clonal mutations (or VRF50(Rx))' were there for each sample?

We now give the absolute no of fixed mutations among ramets, and added one column in the Supplementary Data1, named as "N_fixed". $N_fixed = VRF50 \times 2$, assuming that the distribution is symmetric around $I=0.5$.

What was the sequencing depth for each sample?

The overall sequencing depth was between 75x and 900x for the calibration samples, and 38x-109x for the actual clonal data (see also updated Supplementary Data Table 1).

Moreover, in order to meet our stringent criteria for fixed SNPs, the following conditions had to be met for our parameter VRF50. VRF50(X1, X2) was calculated as the number of SNPs meeting the following criteria: 1) the coverage of X1 ≥ 12 ; 2) the coverage of X2 ≥ 23 ; 3) the variant read frequency of X1 ≤ 0.01 ; 4) the variant read frequency of X2 ≥ 0.50 .

How accurate were the mutation calls?

Every effort was taken to assure that SNP calls were accurate.

- 1) We used very stringent criteria for SNP mapping (see above), including a minimal coverage of any target site to be called at >23 .
- 2) we examined the difference among somatic SNPs called from the oldest Finnish genet, as function of two sequencing depths (80x, 1370x) and platforms (new Supplementary Table 1), and found little effect
- 3) we examined the effects of read coverage by subsampling the ultra-high coverage data of the 4-yr old calibration clones (new Supplementary Table 2), and found little effect
- 4) we compared different mutation callers for asexually derived SNPs (Strelka2 /Mutect2), we now added this comparison as a new Supplementary Table 3, and found general congruence
- 5) We also note that an earlier study determined a 100% accuracy of fixed mutation calls using an independent, restriction enzyme-based method (Yu et al. 2020 Nat Evol Ecol).

Are there any structural events or large-scale gene conversions that may change the clonal mutation numbers?

Structural variation is possibly another source of somatically generated variation. Using the package CNVnator for the most extensively sampled clone at Ängsö (Finland), we only found one structural variation, i.e., a ~200 bp indel, in the oldest Finnish clone (Yu et al. 2020). Taken together, structural variation does not seem to be as prominent as in other plant examples.

(5) The main figures are beautiful, but many of them are 'cartoons', simulation data, or processed data that do not allow validation of their findings or 'critical reading' of the manuscript. Figures 1 and 2 are just illustrations of the 'models'. Figure 3 includes the simulation study rather than real-world observations. Figures 4c-4d are 'processed' data.

As this is an interdisciplinary paper, addressing a more general audience, we were keen that the modelling results and underlying biological processes would be comprehensible to non-specialist readers. Thus, we felt that figures 1&2 were vital to facilitate this. Simulations as shown in Fig 3 are empirical results of models which are key to introduce the generic somatic genetic clock.

(6) To calibrate their model, they used sequencing data from a few samples (5 samples?) from two different time points (4 years and 17 years; Figure 4a). Although the authors showed the linear regression model, I think it is too crude, and the authors need more samples (observations).

It is always true that more samples / time points are better. Here, we have had the unique opportunity to have 2 temporal calibration points, one of which was long-term cultivated to a remarkable age of 17 yrs, in a clonal organism that does not leave traces of old biomass behind that permit later ageing (i.e. trees, corals). The regression line not only explains a substantial part of the variance (adjusted $R^2 = 0.95$), but we also note that the intercept is slightly negative, which is to be expected when we assume a short lag phase at years 0-3 when few fixed mutations accumulate.

We expanded the relevant section in the ms, L226-235, which now reads: “In order to verify that our data could be used to accurately calibrate the clock, we recreated the sampling strategy for both time points, i.e. 4 and 17 yrs, by simulation and estimated the accumulation rate of fixed SoGV (see Fig. 3c and Supplementary Fig. 12b). Considering data from one hundred simulations for each parameter setting, we observe similar estimated rates in all cases. The difference between the mean estimated rate and “true” rate was between 0.1% and 8%, where the maximum difference is for the most extreme case ($N=12$, $N_0=6$, $r=3/\text{yr}$). The standard deviation for each parameter setting ranged between 0.10 and 0.15 (mutations per year). As this was similar in magnitude to other sources of error, we consider that our data can safely be used for calibration. Thus, we consider that our calibration genets with known ages of 4-yr

and 17-yr can be safely used for calibration. However, increasing the number of samples would likely reduce the error resulting from sampling."

(6-1). How many mutations were detected from each of the samples with known ages?

We now give absolute nos of fixed mutation per year, standardized by the mappable genome size (Supplementary Data1)

(7) The authors suggest that some samples from Estonia, Norway, and Finland are several hundred years old. How many mutations were detected in the samples? If the linear regression model (in Figure 4a) is applied to those samples, what are the estimated ages of the samples? What's the age difference between the linear regression model and the author's more sophisticated statistical model?

The number of mutations for each sample, standardized by the mappable genome size, have now been added to Supplementary Data1. As the reviewer suggests, the linear model is applied to those samples, and we have improved the clarity of our ms in that respect. The respective passage was added (see verbatim two questions above), L222.

(8) How does the absolute mutation rate impact the age estimation? What were the mutational signatures (patterns) of the genomic mutations?

This is a very good suggestion. We have added a comparison of the mutational spectra of both, germline and clonal SNPs as new Supplementary Figure 16a,b. Basically, there is little evidence for a deviation among both mutational types, while the general pattern with a predominance of G:C A:T transitions is typical also for other plants.

A section has been added to the ms that reads " We also examined the mutational spectra of the 4 oldest genets detected here to six sexually segregating North Atlantic populations to identify possible differences between germline and asexually generated, mitotic mutations (Supplementary Figure 16). Both mutational patterns show the well-known predominance of transitions over transversions in plants {Ossowski, 2010 #4293}. In particular, G:C->A:T transitions contribute (mean±1SD; n=6 and 4, respectively) 0.61±0.04 and 0.66±0.03% of all SNPs in asexual vs. sexual populations, respectively, regardless of their trinucleotide context (see Supplementary Figures 16a,b)."

28(9) minor point; Fig 1c should be updated. The current version seems that the mutation M3 is subclonal to the mutation M2, and the M4 is subclonal to M3.

Thank you for pointing this out, the figure has been modified accordingly.

Decision Letter, first revision:

21st March 2024

Dear Thorsten,

Thank you for submitting your revised manuscript "A somatic genetic clock for clonal species" (NATECOLEVOL-23112883A). It has now been seen again by the original reviewers and their comments are below. The reviewers find that the paper has improved in revision, and therefore we'll be happy in principle to publish it in Nature Ecology & Evolution, pending minor revisions to comply with our editorial and formatting guidelines.

[REDACTED]

Reviewer #1 (Remarks to the Author):

The authors addressed my concerns well. I have no further issues.

Reviewer #2 (Remarks to the Author):

I thank the authors for their careful consideration of the comments and am satisfied that they have

29addressed them. I have no further comments.

Reviewer #4 (Remarks to the Author):

The authors have addressed most of my questions. The revised manuscript seems to be suitable for publication.

Our ref: NATECOLEVOL-23112883B

10th April 2024

Dear Dr. Reusch,

Thank you for your patience as we've prepared the guidelines for final submission of your Nature Ecology & Evolution manuscript, "A somatic genetic clock for clonal species" (NATECOLEVOL-23112883B). Please carefully follow the step-by-step instructions provided in the attached file, and add a response in each row of the table to indicate the changes that you have made. Please also check and comment on any additional marked-up edits we have proposed within the text. Ensuring that each point is addressed will help to ensure that your revised manuscript can be swiftly handed over to our production team.

****We would like to start working on your revised paper, with all of the requested files and forms, as soon as possible (preferably within two weeks). Please get in contact with us immediately if you anticipate it taking more than two weeks to submit these revised files.****

In recognition of the time and expertise our reviewers provide to Nature Ecology & Evolution's editorial

30process, we would like to formally acknowledge their contribution to the external peer review of your manuscript entitled "A somatic genetic clock for clonal species". For those reviewers who give their assent, we will be publishing their names alongside the published article.

Nature Ecology & Evolution offers a Transparent Peer Review option for new original research manuscripts submitted after December 1st, 2019. As part of this initiative, we encourage our authors to support increased transparency into the peer review process by agreeing to have the reviewer comments, author rebuttal letters, and editorial decision letters published as a Supplementary item. When you submit your final files please clearly state in your cover letter whether or not you would like to participate in this initiative. Please note that failure to state your preference will result in delays in accepting your manuscript for publication.

Cover suggestions

We welcome submissions of artwork for consideration for our cover. For more information, please see our guide for cover artwork.

Nature Ecology & Evolution has now transitioned to a unified Rights Collection system which will allow our Author Services team to quickly and easily collect the rights and permissions required to publish your work. Approximately 10 days after your paper is formally accepted, you will receive an email in providing you with a link to complete the grant of rights. If your paper is eligible for Open Access, our Author Services team will also be in touch regarding any additional information that may be required to arrange payment for your article.

Please note that *Nature Ecology & Evolution* is a Transformative Journal (TJ). Authors may publish their research with us through the traditional subscription access route or make their paper immediately open access through payment of an article-processing charge (APC). Authors will not be required to make a final decision about access to their article until it has been accepted. Find out more about Transformative Journals

Authors may need to take specific actions to achieve compliance with funder and institutional open access mandates. If your research is supported by a funder that requires immediate open access (e.g. according to Plan S principles) then you should select the gold OA route, and we will direct you to the compliant route where possible. For authors selecting the subscription publication route, the journal's standard licensing terms will need to be accepted, including <https://www.nature.com/nature-portfolio/editorial-policies/self-archiving-and-license-to-publish>. Those licensing terms will supersede any other terms that the author or any third party may assert apply to any version of the manuscript.

Please use the following link for uploading these materials:
[REDACTED]

[REDACTED]

Final Decision Letter:

7th May 2024

Dear Thorsten,

We are pleased to inform you that your Article entitled "A somatic genetic clock for clonal species", has now been accepted for publication in Nature Ecology & Evolution.

Over the next few weeks, your paper will be copyedited to ensure that it conforms to Nature Ecology and Evolution style. Once your paper is typeset, you will receive an email with a link to choose the appropriate publishing options for your paper and our Author Services team will be in touch regarding any additional information that may be required

Due to the importance of these deadlines, we ask you please us know now whether you will be difficult to contact over the next month. If this is the case, we ask you provide us with the contact information (email, phone and fax) of someone who will be able to check the proofs on your behalf, and who will be available to address any last-minute problems . Once your paper has been scheduled for online publication, the Nature press office will be in touch to confirm the details.

Acceptance of your manuscript is conditional on all authors' agreement with our publication policies (see www.nature.com/authors/policies/index.html). In particular your manuscript must not be published elsewhere and there must be no announcement of the work to any media outlet until the publication date (the day on which it is uploaded onto our web site).

32Please note that *Nature Ecology & Evolution* is a Transformative Journal (TJ). Authors may publish their research with us through the traditional subscription access route or make their paper immediately open access through payment of an article-processing charge (APC). Authors will not be required to make a final decision about access to their article until it has been accepted. Find out more about Transformative Journals

Authors may need to take specific actions to achieve compliance with funder and institutional open access mandates. If your research is supported by a funder that requires immediate open access (e.g. according to Plan S principles) then you should select the gold OA route, and we will direct you to the compliant route where possible. For authors selecting the subscription publication route, the journal's standard licensing terms will need to be accepted, including [a href="https://www.nature.com/nature-portfolio/editorial-policies/self-archiving-and-license-to-publish"](https://www.nature.com/nature-portfolio/editorial-policies/self-archiving-and-license-to-publish). Those licensing terms will supersede any other terms that the author or any third party may assert apply to any version of the manuscript.

We welcome the submission of potential cover material (including a short caption of around 40 words) related to your manuscript; suggestions should be sent to Nature Ecology & Evolution as electronic files (the image should be 300 dpi at 210 x 297 mm in either TIFF or JPEG format). Please note that such pictures should be selected more for their aesthetic appeal than for their scientific content, and that colour images work better than black and white or grayscale images. Please do not try to design a cover with the Nature Ecology & Evolution logo etc., and please do not submit composites of images related to your work. I am sure you will understand that we cannot make any promise as to whether any of your suggestions might be selected for the cover of the journal.

You can generate the link yourself when you receive your article DOI by entering it

33here: <http://authors.springernature.com/share>.

[REDACTED]

P.S. Click on the following link if you would like to recommend Nature Ecology & Evolution to your librarian <http://www.nature.com/subscriptions/recommend.html#forms>

** Visit the Springer Nature Editorial and Publishing website at www.springernature.com/editorial-and-publishing-jobs for more information about our career opportunities. If you have any questions please click here.**